# MM-RLHF: The Next Step Forward in Multimodal LLM Alignment

Yi-Fan Zhang [1]  Tao Yu [1]  Haochen Tian [1]  Chaoyou Fu [2]  Peiyan Li [1]  Jianshu Zeng [1]  Wulin Xie [1]  Yang Shi [1]
Huanyu Zhang [1]  Junkang Wu [1]  Xue Wang [1]  Yibo Hu [1]  Bin Wen [3]  Tingting Gao [3]  Zhang Zhang [1]  Fan Yang [3]
Di Zhang [3]  Liang Wang [1]  Rong Jin [1]

## Abstract

Existing efforts to align multimodal large language models (MLLMs) with human preferences have only achieved progress in narrow areas, such as hallucination reduction, but remain limited in practical applicability and generalizability. To this end, we introduce MM-RLHF, a dataset containing **120k** fine-grained, human-annotated preference comparison pairs. This dataset represents a substantial advancement over existing resources, offering superior size, diversity, annotation granularity, and quality. Leveraging this dataset, we propose several key innovations to improve both the quality of reward models and the efficiency of alignment algorithms. Notably, we introduce the **Critique-Based Reward Model**, which generates critiques of candidate texts before assigning scores, offering enhanced interpretability and more informative feedback compared to traditional reward models. Additionally, we propose **Dynamic Reward Scaling**, a method that adjusts the loss weight of each training sample according to the reward signal, thereby optimizing the use of high-quality comparison pairs. Our approach is rigorously evaluated across **10** distinct dimensions, encompassing **27** benchmarks, with results demonstrating significant and consistent improvements in model performance (Figure. 1).

## 1. Introduction

Although Multimodal Large Language Models (MLLMs) have demonstrated remarkable potential in addressing complex tasks that involve the integration of vision, language, and audio, state-of-the-art MLLMs today seldom undergo a rigorous alignment stage (Wang et al., 2024a; Deitke et al., 2024; Chen et al., 2024c; Dai et al., 2024; Agrawal et al., 2024). Typically, these models only progress to the supervised fine-tuning (SFT) stage, leaving critical aspects such as truthfulness, safety, and alignment with human preferences largely unaddressed. While recent efforts have begun to explore MLLM alignment, they often focus on narrow domains, such as mitigating hallucination or enhancing conversational capabilities, which fail to comprehensively improve the model's overall performance and reliability. This raises a critical question:

> *Is alignment with human preferences only capable of enhancing MLLMs in a limited set of tasks?*

In this work, we confidently answer this question with a resounding "No.". We demonstrate that a well-designed alignment pipeline can comprehensively enhance MLLMs along multiple dimensions, including visual perception, reasoning, dialogue, and trustworthiness, thereby significantly broadening their practical applicability. To achieve this, we conduct in-depth investigations into three pivotal areas: data curation, reward modeling, and alignment algorithms.

At first, we introduce **MM-RLHF**, a dataset designed to advance **M**ultimodal **R**einforcement **L**earning from **H**uman **F**eedback (RLHF). The dataset spans three key domains: image understanding, video understanding, and MLLM safety. Constructed through a rigorous pipeline, MM-RLHF ensures high-quality, fine-grained human annotations. Dataset creation process involves the following steps (Figure 2):

- **Data Collection**. We curate a diverse set of multimodal tasks from various sources, totaling 10 million data instances, ensuring broad representation across tasks.

- **Data Selection**. Through rigorous re-sampling, we extract 30k representative queries, ensuring diversity across a wide range of data types, such as real-world scenarios, mathematical reasoning, chart understanding, and other practical domains (Figure. 3).

- **Model Response Generation**. We utilize state-of-the-

[1]Institute of Automation, School of Artificial Intelligence, University of Chinese Academy of Sciences (UCAS), Beijing, China [2]Nanjing University (NJU), Nanjing, China [3]KuaiShou, Beijing, China. Correspondence to: Chaoyou Fu <bradyfu24@gmail.com>, Bin Wen <wenbin@kuaishou.com>, Zhang Zhang <zhangzhang@casia.edu.cn>.

*Proceedings of the $42^{nd}$ International Conference on Machine Learning*, Vancouver, Canada. PMLR 267, 2025. Copyright 2025 by the author(s).

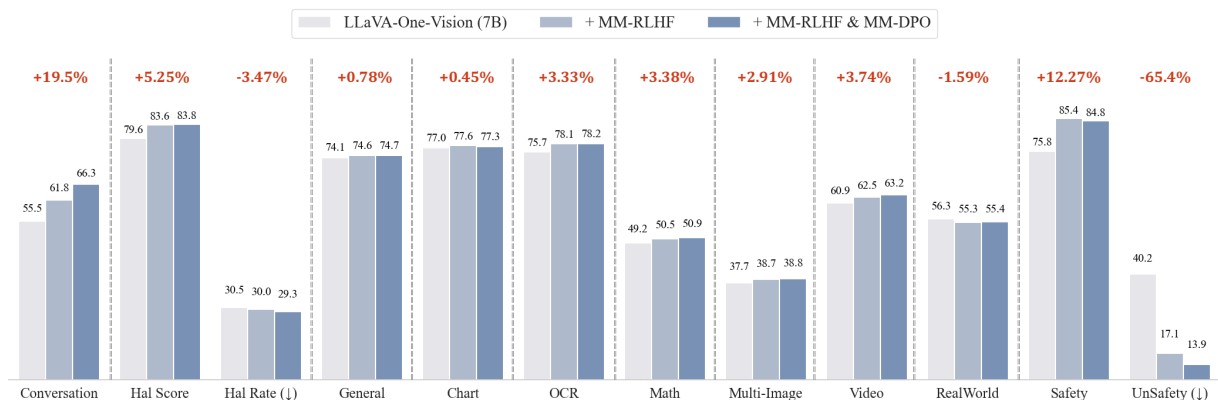

Figure 1. Performance gains achieved through alignment training on MM-RLHF and our alignment algorithm (MM-DPO), highlighting significant improvements across various tasks and metrics.

art models, such as Claude 3.5-Sonnet and Qwen2-VL-72B, to generate responses for various tasks.

- **Fine-grained Human Annotation**. We employ a meticulous annotation process, involving over 50 annotators over two months, to score, rank, and provide textual explanations for responses. This results in more than 120k high-quality ranked comparison pairs.

Compared to existing datasets, MM-RLHF significantly advances in diversity, response quality, and annotation granularity, providing a robust foundation for MLLM alignment.

Building on the MM-RLHF dataset, we investigate how human-annotated data can enhance MLLM alignment, with a focus on reward modeling and training optimization. Recognizing the pivotal role of reward models in providing feedback signals to guide the alignment process, we propose a **Critique-Based Reward Model** (Figure 4). Traditional reward models, which output scalar values, often lack interpretability, while directly using MLLMs as reward models place high demands on their instruction-following capabilities, limiting their practicality. To address these limitations, we first transform concise human annotations into detailed, model-friendly formats using MLLMs. These enriched annotations serve as learning targets, guiding the reward model to first generate critiques and then assign scores based on the critiques. This approach enables the model to provide fine-grained scoring explanations, significantly enhancing the quality and interpretability of the reward signals. **MM-RLHF-Reward-7B** achieves SOTA performance on several benchmarks, outperforming several 72B-scale models.

Building on this high-quality reward model, we introduce **Dynamic Reward Scaling** within the Direct Preference Optimization (DPO) framework. Traditional DPO methods (Amini et al., 2024) use a fixed training weight for all human-preferred and non-preferred training pairs. In contrast, Dynamic Reward Scaling calculates a reward margin for each comparison pair using MM-RLHF-Reward-7B. During training, it assigns higher weights to comparison pairs with larger reward margins. This ensures that the most informative samples have a stronger influence on model updates. As a result, the training process becomes more efficient, leading to improved model performance.

Finally, to rigorously evaluate our approach, we construct two specialized benchmarks. The first, **MM-RLHF-RewardBench**, is sampled from our dataset and consists of meticulously human-annotated data for evaluating reward models. The second, **MM-RLHF-SafetyBench**, is curated and filtered from existing benchmarks and focuses on safety-related tasks, including privacy protection, adversarial attacks, jailbreaking, and harmful content detection.

We conduct extensive evaluations across ten key dimensions, covering 27 benchmarks. The results demonstrate that our training algorithm, combined with the high-quality MM-RLHF dataset, leads to significant improvements in model performance. Specifically, models fine-tuned with our approach achieve an average 11% gain in conversational abilities and a 57% reduction in unsafe behavior. The integration of our reward model further amplifies these gains, highlighting the effectiveness of our alignment algorithm.

## 2. MM-RLHF-Dataset

In this section, we outline the construction of MM-RLHF, as illustrated in Figure 2. This includes data collection, data filtering, and human annotation.

### 2.1. Data Collection

Our goal is to construct a comprehensive post-training dataset that covers a wide range of task types. To achieve this, we categorize tasks into three main domains: image

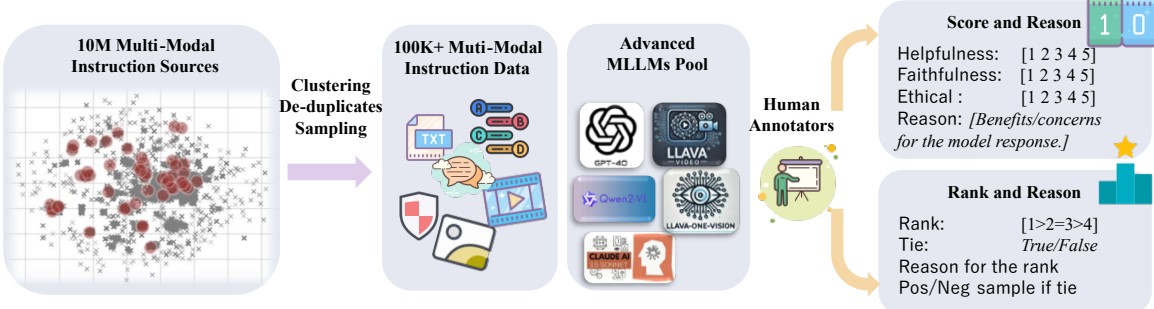

Figure 2. **MM-RLHF Construction Pipeline**. *(1) Data Collection and Cleaning*: Starting with 10 million instruction samples, we cluster data based on image similarity, and uniformly sample across diverse categories. This results in a diverse dataset covering image-based Q&A (e.g., multiple-choice, dialogues, and safety-related questions) and video Q&A formats. *(2) Response Generation*: We leverage state-of-the-art models, including GPT-4o and Qwen2-VL-72B, to generate responses. *(3) Human Annotation*: We conduct comprehensive manual annotation across nine categories, including scoring, ranking, and textual explanations, ensuring fine-grained evaluation.

understanding, video understanding, and multimodal safety.

For **image understanding**, we integrate data from multiple sources, including LLaVA-OV (Li et al., 2024c), VLfeedback (Li et al., 2023d), LLaVA-RLHF (Sun et al., 2023b), lrv-instruction (Liu et al., 2023a), and Unimm-Chat (Yu et al., 2023). Since some datasets contain multi-turn dialogues, which are less suitable for response generation, we decompose them into single-turn dialogues. This process yields over 10 million dialogue samples, covering tasks such as conversation, safety, multiple-choice questions, captions, and commonsense reasoning.

For **video understanding**, the primary data source is SharedGPT-4 video (Chen et al., 2024b).

For **safety**, data is primarily derived from VLGuard (Zong et al., 2024) and self-constructed content. VLGuard contains over 2,000 harmful samples, while additional red teaming, safety, and robustness data are included. The pipeline for constructing safety data is detailed in Appendix C.1.

## 2.2. Data Filtering and Model Response Generation

The core goal of data filtering is to reduce the number of samples while maintaining the diversity of the original dataset. To achieve this, the following strategies are adopted:

**Predefined Sampling Weights.** For image understanding tasks, we define three categories based on the nature of the questions and the length of model responses: 1. *Multiple-choice questions (MCQ)*: *questions with options such as A, B, C, or D.* These tasks include visual question answering, mathematics, OCR, and icon recognition, focusing on the model's reasoning and visual perception abilities. 2. *Long-text questions*: *questions for which GPT-4o generates responses exceeding 128 characters.* These typically involve detailed captions or complex descriptions, testing the model's conversational and descriptive capabilities. 3. *Short-text questions*: *questions for which GPT-4o generates*

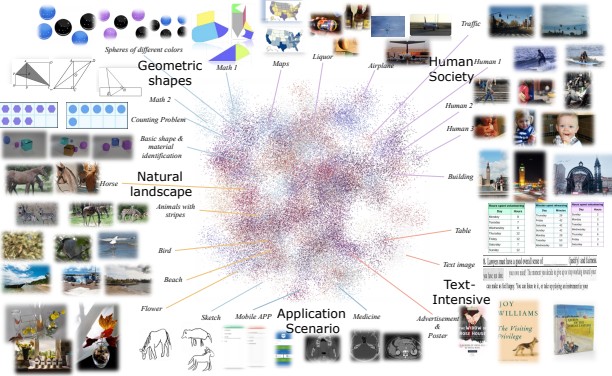

Figure 3. **Re-Sampling results from the clustering process.** Due to the large total number of samples, the clustered and deduplicated results contain a rich diversity of categories. Selected samples include topics such as mathematics, daily life, natural scenes, medicine, electronic technology, and OCR scenarios, showcasing a variety of problem-image pairs. The 2D features are obtained via UMAP dimensionality reduction.

*responses shorter than 128 characters.* These require concise answers, often involving simple image analysis, and represent a broader range of task types.

The initial distribution of these three types in the image understanding dataset is highly imbalanced, with proportions of 12.17% (Long), 83.68% (Short), and 4.14% (MCQ). To align with diversity goals, we adjust the sampling ratio to 4:5:1, reducing disparities among task types while maintaining a dominance of comprehensive samples. [1]

**Cluster-based Sampling.** Text deduplication is not performed because many questions, while similar in text, are paired with different images, leading to substantially different outcomes—an intrinsic characteristic of multimodal

---

[1]For video understanding and safety tasks, MCQ samples are fewer. After classifying into Long and Short types, the differences are minimal, so no additional adjustments are made.

Table 1. Dataset Composition Statistics

| Image | | | Safety | Video | Total |
|---|---|---|---|---|---|
| Long | Short | MCQ | | | |
| 9575 | 12063 | 2125 | 1999 | 4235 | 29997 |

data. Instead, we encode all images using CLIP [2], and for videos, we use the feature of the first frame as a representative. We then apply KNN clustering with 100 cluster centers and randomly sample $N$ instances from each cluster. The value of $N$ is determined to satisfy the predefined sampling ratios, ensuring a balanced representation of task diversity.

**Data Statistics** is summarized in Table 1, and the visualization of the clustering results is shown in Figure 3, demonstrating the rich diversity of data categories.

**Model Response Generation.** To generate high-quality responses, we select state-of-the-art models from both open-source and closed-source domains. For image understanding and safety-related tasks, we use Qwen2-VL-72B (Wang et al., 2024a), LLaVA-OV-72B (Li et al., 2024c), GPT-4o[3], and Claude 3.5-sonnet[4]. For video understanding tasks, we employ GPT-4o, LLaVA-Video-72B (Zhang et al., 2024b), and Qwen2-VL-72B (Wang et al., 2024a). These models are chosen for their advanced capabilities and performance, ensuring a comprehensive evaluation of leading solutions in multimodal understanding.

### 2.3. Annotation

The annotation process follows rigorous standards to ensure comprehensive and fine-grained evaluations of MLLM responses. Detailed standards are provided in Appendix B, and the scoring and annotation structure are illustrated in Figure 2. Additionally, we design a web UI to streamline the annotation process, as shown in Figure 6.

Compared to prior work, our annotation approach introduces two significant advantages: **richness** and **granularity**. First, the evaluation incorporates three core dimensions—*Helpfulness*, *Faithfulness*, and *Ethical Considerations*—to comprehensively capture model performance. *Helpfulness* ensures that responses are relevant and provide meaningful assistance aligned with the user's intent. *Faithfulness* evaluates the accuracy of responses in describing visual elements, such as objects, relationships, and attributes, ensuring alignment with the ground truth while avoiding hallucinated content. *Ethical Considerations* assess adherence to ethical principles, including safety, privacy, fairness, and

---

[2] https://huggingface.co/openai/clip-vit-base-patch32
[3] https://openai.com/index/hello-gpt-4o/
[4] https://www.anthropic.com/news/claude-3-5-sonnet

harm avoidance, ensuring responses are free from harmful or biased content. Annotators score each dimension while documenting the reasoning behind their assessments, adding valuable context for understanding model performance.

Second, annotators are required to assign an **overall ranking** to the responses, along with justifications for their rankings. This ranking mechanism provides a transparent and nuanced comparison of model outputs. Additionally, innovative strategies are employed to enhance data quality:

**Constructing Positive Samples for Poor Quality Ties.** When multiple responses are equally poor, annotators provide correct answers to create positive examples. This ensures challenging samples contribute to the training dataset, addressing issues that no valid model responses exist.

**Constructing Negative Samples for High-Quality Ties.** When multiple responses are of equally high quality, annotators introduce deliberate errors to create negative samples. This prevents ties from reducing the utility of the data and allows for more efficient use in training.

We employ over 50 annotators supervised by 8 multimodal experts to ensure high-quality annotations, addressing the fine-grained requirements of MLLM alignment tasks. While human annotation incurs significant costs, it offers substantial advantages over machine annotation, particularly in capturing nuanced perceptual differences and providing professional-grade reasoning. For a detailed comparison of human and machine annotation, see Appendix D.

## 3. MM-RLHF-Reward Model

In this section, we explore how to train a high-quality reward model using MM-RLHF, to provide a robust supervision signal for subsequent model alignment. The reward model is designed to combine critique generation and scoring (Figure 4), ensuring a comprehensive evaluation process.

### 3.1. Background of Standard Reward Models

Reward models are a key component for aligning model outputs with human preferences. Typically, a reward model starts with a pretrained LLM $\phi$, where the LLM head $h_l$ is replaced with a linear reward head $l_r$, enabling the model to output a scalar reward. These models are trained using pairwise comparisons. Given a query $\mathbf{x}$, a preferred response $y_w$, and a less preferred response $y_l$, the reward model is optimized to assign higher rewards to preferred responses:

$$\ell_{\text{Reward}}(\theta) = \mathbb{E}_{\mathbf{x}, y_w, y_l} \left[ -\log \sigma \Big( r(y_w|\mathbf{x}) - r(y_l|\mathbf{x}) \Big) \right], \quad (1)$$

where $r(y|\mathbf{x})$ is the reward and $\sigma$ is the sigmoid function.

Standard reward models face significant limitations. First, they fail to fully utilize the rich feedback provided by high-

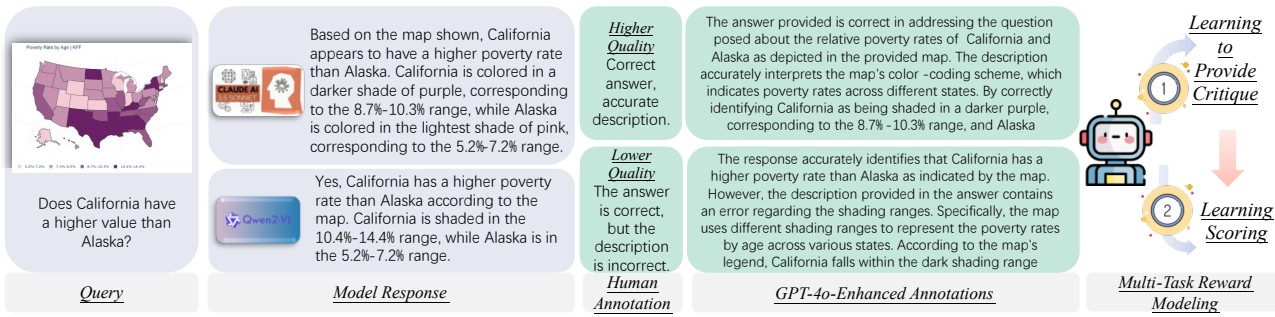

Figure 4. **Illustration of the multi-task reward model training process.** The process begins with a user query and corresponding model responses, which are ranked and annotated by humans. Human annotations are expanded using GPT-4o to provide enhanced rationales. The reward model is trained with two objectives: (1) *Learning to Provide Critique*, where the model learns to provide detailed critiques and evaluations for model responses, and (2) *Learning Scoring*, where the model learns to assign scores based on the model response and critique. The integration of these tasks ensures a robust evaluation framework for improving model outputs.

quality human annotations, such as textual explanations and nuanced reasoning. Second, scalar rewards lack transparency, making it difficult for humans to understand how the reward is generated. These challenges highlight the need for a more interpretable and robust reward model that leverages critiques as intermediate reasoning steps.

### 3.2. Critique-Based Reward Model Training

**Extending to Critique-Based Training.** To overcome the limitations of traditional reward models, we propose a critique-based training framework: the model first generates a critique $c$ conditioned on the query $x$. This critique serves as an intermediate reasoning step, providing context for scoring responses. The critique-based reward model comprises two components: **1. Critique Head** ($h_l$): Generates critiques $c_w$ and $c_l$ for the preferred ($y_w$) and less preferred ($y_l$) responses, respectively, based on the query $x$. **2. Scoring Head** ($h_r$): Assigns scalar rewards based on the generated critiques, enabling more fine-grained evaluation.

**Learning to Provide Critique from Enhanced Annotation.** The $h_l$ is trained to align with human-provided annotations. The loss function for critique generation is:

$$\ell_{\text{Critique}}(\theta) = \mathbb{E}_{x,y,c}\Big[-\sum_{t=1}^{|c|} \log \pi_\theta(c_t|c_{<t}, x, y)\Big], \quad (2)$$

where $c_t$ is the $t$-th token in the critique $c$, $c_{<t}$ denotes the tokens preceding $c_t$, and $\pi_\theta(c_t|c_{<t}, x, y)$ is the probability of token $c_t$ given its context, query $x$, and response $y$.

However, as shown in Figure 4, while human-provided scoring reasons are highly accurate, they tend to be concise. Directly using these concise annotations as training targets for the reward model's language head does not yield significant performance improvements. To address this issue, we use GPT-4o to augment the human annotations by adding more

details and improving the fluency of the critiques. These enhanced scoring reasons are then used as the training targets for the language head. To prevent GPT-4o from introducing hallucinated content or irrelevant analysis, we impose strict constraints in the prompt (Table. 7), to ensure the model only expands on the original content without introducing speculative or uncertain information.

**Scoring Loss with Teacher-Forcing.** $h_r$ computes scalar rewards based on the query $x$, response $y$, and critique $c$. During training, we adopt a teacher-forcing strategy, where the scoring head uses ground truth critiques instead of critiques generated by itself. This avoids potential noise from model-generated critiques in the early stages of training. The scoring loss is defined as:

$$\ell_{\text{Score}}(\theta) = \mathbb{E}_{x,y_w,y_l}\Big[-\log\sigma\Big(r(x,y_w,c_w) - r(x,y_l,c_l)\Big)\Big], \quad (3)$$

where: $c_w$ and $c_l$ are the ground truth critiques for the responses $y_w, y_l$, respectively, $r(x, y, c)$ is the reward score computed from $x$, $y$, and $c$.

**Joint Training Objective.** The overall training objective combines the critique generation loss and the scoring loss: $\ell_{\text{Total}}(\theta) = \ell_{\text{Critique}}(\theta) + \ell_{\text{Score}}(\theta)$.

**Inference.** During inference, the critique head ($h_l$) generates a critique $c$ conditioned on the query $x$ and response $y$. The scoring head ($h_r$) then uses $x$, $y$, and the generated critique $c$ to compute the final reward score $r(x, y, c)$. This two-step process mirrors the human evaluation process by explicitly reasoning about critiques before scoring.

In Section E, we discuss the differences and connections between the **MM-RLHF-Reward Model** and existing works.

**MM-RLHF-RewardBench.** To evaluate the effectiveness of the signals provided by our reward model in guiding subsequent model training, we randomly sample 10 examples

from each category of the MM-RLHF dataset to create a test set. Each example includes multiple model responses and their corresponding rankings, enabling the generation of several comparison pairs. This results in a total of 170 pairs for evaluation. We design two evaluation metrics: 1. *Traditional Accuracy (ACC)*: Measures the proportion of cases where the model correctly identifies the preferred response. 2. *ACC+*: Measures the proportion of cases where the model correctly ranks all response pairs for a given sample. This metric emphasizes the model's ability to handle challenging cases, such as those with small ranking differences or hard-to-distinguish pairs.

## 4. MM-DPO

In this section, we propose MM-DPO, an extension of the traditional DPO framework. MM-DPO introduces Dynamic Reward Scaling, which dynamically adjusts the update strength based on the confidence of training pairs, ensuring effective utilization of high-quality samples while mitigating the impact of noisy or low-confidence data.

### 4.1. Background: Direct Preference Optimization

The DPO framework is a preference-based learning method that optimizes model parameters $\theta$ by aligning model outputs with human preferences. Given a query $\mathbf{x}$ and corresponding responses $y_w$ (positive) and $y_l$ (negative), the DPO loss is defined as: $\ell_{\text{DPO}}(\theta) = \mathbb{E}_{\mathbf{x}, y_w, y_l} \Big[ -\log \sigma \Big( \beta \Big( \log \frac{\pi_\theta(y_w|\mathbf{x})}{\pi_{\text{ref}}(y_w|\mathbf{x})} - \log \frac{\pi_\theta(y_l|\mathbf{x})}{\pi_{\text{ref}}(y_l|\mathbf{x})} \Big) \Big) \Big]$, where $\pi_\theta$ is the optimal policy, $\pi_{\text{ref}}$ is the reference policy, $\beta$ is a scaling factor, and $\sigma(\cdot)$ is the sigmoid function (Amini et al., 2024). Traditional DPO treats all training pairs equally, regardless of their quality differences. This uniform scaling fails to prioritize high-quality pairs with clear preference distinctions, leading to inefficient use of informative samples and suboptimal optimization.

### 4.2. MM-DPO: Key Contributions and Improvements

**Training on all possible comparison pairs instead of the hardest pairs.** Unlike many recent MLLM alignment approaches that prioritize training on the hardest comparison pairs, MM-DPO incorporates all possible comparison pairs for a single query into the training process. Specifically, for any query with multiple responses, every response pair with differing ranks is treated as a valid comparison pair. This comprehensive approach captures more nuanced ranking information, allowing the model to learn from a broader set of preferences. However, this strategy also introduces a challenge: pairs involving responses with similar ranks (e.g., rank 3 and rank 4) often have lower reward margins compared to pairs with more distinct rankings (e.g., rank 1 and rank 4). Treating all pairs equally exacerbates the issue

of uniform scaling and underutilizes the high-confidence information contained in larger reward margins. To address this, MM-DPO introduces Dynamic Reward Scaling, which dynamically adjusts the update strength based on the reward margin to prioritize high-confidence training pairs.

**Definition of Dynamic Reward Scaling.** Reward models can naturally provide a pairwise reward margin, which serves as a straightforward signal for scaling. However, two critical aspects must be addressed: (1) ensuring the signal quality is sufficiently high, and (2) bounding the signal to prevent overly aggressive updates that destabilize training.

Regarding the first aspect, our experiments reveal that publicly available models, such as GPT-4o and LLaVA-Critic, perform inadequately in scoring our dataset. Conversely, our trained MM-RLHF-Reward-7B model surpasses several publicly available 72B models, offering a reliable and robust reward signal. We use this model to compute the reward margin: $\delta = r(y_w) - r(y_l)$, where $r(y_w)$ and $r(y_l)$ are the scores assigned to the positive and negative samples.

For the second factor, we control the scaling factor $\beta(\delta)$ using the following formulation: $\beta(\delta) = \beta_{\text{ori}} \Big( 1 + w \big( 1 - e^{-k\delta} \big) \Big)$, where $\beta_{\text{ori}}$ is the initial default scaling factor, $w$ is a parameter balancing the dynamic component's contribution, and $k$ is a hyperparameter that adjusts $\beta(\delta)$'s sensitivity to changes in $\delta$. The function $1 - e^{-k\delta}$ is bounded between $[0, 1]$, and a smaller $k$ value keeps most $\beta(\delta)$ values near $\beta_{\text{ori}}$, with slow growth as $\delta$ increases. In contrast, a larger $k$ makes $\beta(\delta)$ highly responsive to changes in $\delta$, quickly reaching its maximum. To avoid aggressive updates, we constrain $\beta(\delta)$ within $[\beta_{\text{ori}}, (1 + w)\beta_{\text{ori}}]$. Overall, Dynamic Reward Scaling significantly enhances MM-DPO by leveraging high-quality reward signals and tailoring optimization steps to the confidence level of training pairs. This results in improved robustness, efficiency, and effectiveness of the framework. We discuss the similarities and differences between our approach and existing methods in Appendix F.

## 5. Experiments

In this section, we evaluate our data and algorithms on 10 tasks across 20+ benchmarks. The key findings are:

1. Alignment training on the **MM-RLHF** dataset consistently improves performance across nearly all benchmarks for various baselines. The integration of reward signals in MM-DPO further amplifies these improvements, demonstrating the effectiveness of our approach.

2. The **MM-RLHF-Reward-7B** model achieves state-of-the-art performance on reward model benchmarks among open-source models, surpassing even several 72B models. This highlights the efficiency and scalability of our method.

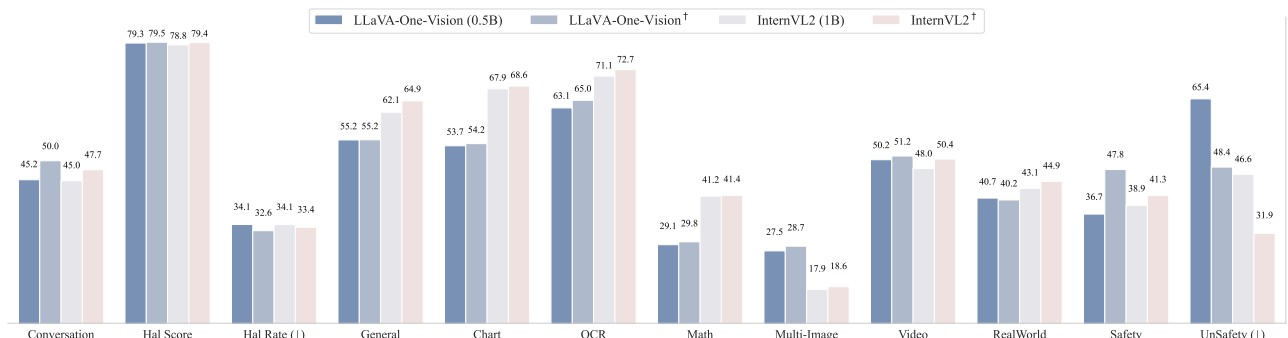

*Figure 5.* Performance improvements achieved through alignment training. † indicates the use of our dataset and alignment algorithm.

*Table 2.* **Performance comparison across metrics and methods on MM-RLHF-RewardBench**. *MM-RLHF-Reward (w/o. Task 1)* represents training the LLaVA-OV-7B model to score pair-wise samples while excluding Task 1. *MM-RLHF-Reward (w/o. enhanced annotations)* involves learning human-provided annotations, followed by scoring. *MM-RLHF-Reward (inference w. GT annotation)* uses ground truth annotations during inference.

| Method | LLaVA-OV-7B | | LlaVA-Critic (Pointwise) | | LlaVA-Critic (Pairwise) | | GPT-4o | | MM-RLHF-Reward (w/o. Task 1) | | MM-RLHF-Reward (w/o. enhanced annotations) | | MM-RLHF-Reward | | MM-RLHF-Reward (inference w. GT annotation) | |
|---|---|---|---|---|---|---|---|---|---|---|---|---|---|---|---|---|
| Metric | ACC | ACC+ | ACC | ACC+ | ACC | ACC+ | ACC | ACC+ | ACC | ACC+ | ACC | ACC+ | ACC | ACC+ | ACC | ACC+ |
| Mcq | 0.14 | 0.00 | 0.38 | 0.10 | 0.23 | 0.00 | 0.69 | 0.20 | 0.90 | 0.80 | 0.83 | 0.70 | 0.93 | 0.70 | **1.00** | **1.00** |
| Long | 0.11 | 0.00 | 0.49 | 0.20 | 0.54 | 0.30 | 0.95 | 0.90 | 0.70 | 0.40 | 0.92 | 0.80 | 1.00 | 1.00 | **1.00** | **1.00** |
| Short | 0.29 | 0.20 | 0.38 | 0.20 | 0.24 | 0.10 | 0.56 | 0.40 | 0.79 | 0.60 | 0.68 | 0.40 | 0.71 | 0.50 | **1.00** | **1.00** |
| Safety | 0.41 | 0.00 | 0.62 | 0.17 | 0.28 | 0.17 | **0.72** | **0.33** | 0.69 | **0.33** | 0.69 | 0.17 | 0.66 | 0.17 | 0.69 | 0.17 |
| Video | 0.32 | 0.10 | 0.40 | 0.20 | 0.52 | 0.20 | 0.80 | 0.60 | 0.70 | 0.60 | 0.80 | 0.60 | **0.92** | 0.80 | **0.92** | **0.90** |
| Overall | 0.24 | 0.07 | 0.45 | 0.17 | 0.35 | 0.15 | 0.74 | 0.50 | 0.75 | 0.50 | 0.79 | 0.57 | 0.85 | 0.67 | **0.93** | **0.87** |

3. We conduct extensive ablation studies and analyses, such as investigating the importance of critique learning for reward models and the sensitivity to hyperparameters. Additionally, we identify several experimental phenomena that challenge mainstream perspectives, such as the observation that small-scale MLLMs struggle to perform effective self-improvement. Due to space constraints, experimental setups and additional findings are provided in Appendix G.

### 5.1. Evaluation of MM-RLHF and MM-DPO

Figures 1 and 5 illustrate the alignment performance of LLaVA-OV-7B and InternVL-1B using our dataset and alignment algorithm, where the scores for each evaluation dimension are averaged across their respective benchmarks. We also conduct experiments on LLaVA-OV-0.5B, with detailed benchmark results provided in Appendix Table 8 (for understanding tasks) and Table 9 (for safety tasks).

**Significant improvements in conversational ability and safety.** The alignment process of LLaVA-OV-7B leads to substantial improvements in these two aspects without requiring hyperparameter tuning. The average improvement in conversational benchmarks exceeds 15%, while unsafe behaviors are reduced by 65%. Additionally, in WildsVision, the win rate increases by 144%. This suggests that existing MLLMs lack explicit optimization for these dimensions, and our dataset effectively fills this gap.

**Broad enhancements in hallucination, mathematical reasoning, multi-image, and video understanding.** The aligned models also exhibit notable improvements in these areas. Interestingly, despite the lack of dedicated multi-image data in our dataset, the model's performance in multi-image tasks improves significantly. This indicates that the diversity of our alignment data enhances generalization across multiple dimensions.

**Model-specific preferences for data and hyperparameter selection.** Different models exhibit varying performance trends during alignment, with distinct preferences for hyperparameter settings across different benchmarks. For instance, in our training of InternVL-1B, we found that excluding the SFT loss led to better results. Additionally, while InternVL-1B demonstrates significant improvements in general knowledge tasks, its relative enhancement in OCR tasks is less pronounced compared to the LLaVA-OV series. These differences largely stem from variations in the models' pretraining datasets and strategies, necessitating tailored hyperparameter adjustments for optimal alignment.

**Limited gains in high-resolution benchmarks.** The model shows no significant improvement on high-resolution benchmarks, likely because our dataset contains relatively few ultra-high-resolution images. Additionally, our filtering strategy is based on image similarity rather than resolution, meaning the alignment process does not explicitly optimize for high-resolution tasks. As a result, performance gains in

this area remain limited.

**Ablation studies and sensitivity analysis.** To further validate the effectiveness of our approach, we provide detailed ablation studies in the appendix, analyzing the impact of different alignment parameters and the improvements introduced by our dataset and MM-DPO.

## 5.2. Evaluation of MM-RLHF-Reward

In this section, we evaluate the effectiveness of MM-RLHF-Reward and highlight several noteworthy experimental observations. The results are presented in Table 2 and Table 3.

**Existing reward models exhibit significant overfitting.** As shown in Table 2, LLaVA-Critic's performance on MM-RLHF-RewardBench is suboptimal, with a considerable gap compared to GPT-4o. This can likely be attributed to the overfitting of existing reward models to their training data, which predominantly consists of conversational datasets and real-world images. Consequently, while LLaVA-Critic demonstrates notable improvements over its baseline, LLaVA-OV-7B[5], its performance in other categories, such as MCQ, remains limited.

**Closed-source models consistently deliver competitive performance.** Across both Table 2 and Table 3, closed-source models such as GPT-4o demonstrate superior generalization capabilities compared to open-source alternatives, even those with significantly larger parameter sizes. This observation underscores the robustness of closed-source approaches in handling diverse multimodal tasks and maintaining high performance across various metrics.

**MM-RLHF-Reward establishes a new standard for open-source models.** In both benchmarks, MM-RLHF-Reward achieves results comparable to or exceeding GPT-4o, while significantly outperforming most open-source models, such as Qwen2-VL-72B-Instruct. Notably, on our custom benchmark, MM-RLHF-Reward demonstrates a substantial lead over GPT-4o, further justifying its selection as the reward signal for training algorithms. Its robust performance across diverse metrics highlights its effectiveness and adaptability.

**Importance of effective critics in reward modeling.** The results in Table 2 highlight the critical role of effective critics in reward modeling. Training the reward head directly on pairwise datasets yields an ACC+ of around 50%. By incorporating human annotations as learning targets—enabling the model to first learn evaluation reasoning before scoring—ACC+ consistently improves by 5%. However, human annotations alone may not suffice as optimal training targets

---

[5]Both models use identical prompts for tasks such as captioning and long-form dialogue.

*Table 3.* **Performance comparison of our reward model (MM-RLHF-Reward) with existing open-source and private MLLMs.** MM-RLHF-Reward-7B outperforms existing 72B open-source MLLMs and several competitive closed-source models.

| Model | General | Hallucination | Reasoning | Avg |
|---|---|---|---|---|
| VITA-1.5 (Fu et al., 2025) | 18.55 | 8.93 | 22.11 | 16.48 |
| SliME-8B (Zhang et al., 2024c) | 7.23 | 27.09 | 18.6 | 19.04 |
| deepseek-vl2 (Wu et al., 2024b) | 29.70 | 23.80 | 50.90 | 34.80 |
| Phi-3.5-vision-instruct (Abdin et al., 2024) | 28.00 | 22.40 | 56.60 | 35.67 |
| llava-onevision-qwen2-7b-ov (Li et al., 2024c) | 32.20 | 20.10 | 57.10 | 36.47 |
| Molmo-7B-D-0924 (Deitke et al., 2024) | 31.10 | 31.80 | 56.20 | 39.70 |
| Pixtral-12B-2409 (Agrawal et al., 2024) | 35.60 | 25.90 | 59.90 | 40.47 |
| Qwen2-VL-72B-Instruct (Wang et al., 2024a) | 38.10 | 32.80 | 58.00 | 42.97 |
| NVLM-D-72B (Dai et al., 2024) | 38.90 | 31.60 | 62.00 | 44.17 |
| InternVL2-26B (Chen et al., 2024c) | 39.30 | 36.90 | 60.80 | 45.67 |
| *Private models* | | | | |
| GPT-4o-mini (2024-07-18) | 41.70 | 34.50 | 58.20 | 44.80 |
| Claude-3.5-Sonnet (2024-06-22) | 43.40 | 55.00 | 62.30 | 53.57 |
| GPT-4o (2024-08-06) | 49.10 | 67.60 | 70.50 | 62.40 |
| Gemini-1.5-Pro (2024-09-24) | 50.80 | 72.50 | 64.20 | 62.50 |
| *Ours* | | | | |
| MM-RLHF-Reward-7B | 45.04 | 50.45 | 57.55 | 50.15 |

due to their brevity or conversational style. To address this, we enrich human annotations using GPT-4o, significantly enhancing reward model training quality and achieving a 17% improvement in ACC+ over the baseline. Notably, when human annotations are directly used as critics during evaluation (i.e., scoring is based on human-provided evaluations rather than model-generated critiques), both ACC and ACC+ reach approximately 90%, underscoring the pivotal role of critique quality in reward model effectiveness.

**Multiple sampling of critiques does not yield performance gains.** While prior research in LLMs suggests that sampling multiple critiques and averaging their scores can improve performance (Yu et al., 2024d), our experiments show that this approach leads to performance degradation when applied to our model. This is because lowering the sampling temperature occasionally produces inaccurate critiques. Since our model, aligned with human annotations, already generates reasonably accurate critiques, the additional sampling process is not only time-consuming but also counterproductive, negatively impacting performance.

*Table 4.* **Comparison of DPO baselines**, where all models are trained on the MM-RLHF training dataset.

| Method | LLaVA-Wild | OCRBench | DocVQA | MathVista | MMHal |
|---|---|---|---|---|---|
| LLaVA-Ov-7B | 90.70 | 62.30 | 84.34 | 60.10 | 3.22 |
| DPO | 95.20 | 66.30 | 85.72 | 60.90 | 3.88 |
| beta-DPO | 94.40 | 66.50 | 85.44 | 60.00 | 3.44 |
| SimPO | 93.00 | 64.50 | 84.50 | 59.10 | 3.30 |
| LLaVA-Critic | **97.90** | 64.20 | 84.31 | 58.74 | 3.95 |
| MIA-DPO | 95.00 | 66.50 | 85.60 | 60.00 | 3.80 |
| MPO | 95.30 | 66.20 | 85.88 | 61.10 | 3.72 |
| MM-DPO | **97.90** | **69.30** | **86.11** | **61.60** | **4.08** |

## 5.3. Comparison of DPO Baselines and Other Alignment Datasets

We compared several baselines including DPO (Amini et al., 2024), LLaVA-Critic (Xiong et al., 2024), beta-DPO (Wu

*Table 5.* **Comparison of preference datasets**, where all models are trained using the DPO algorithm.

| Dataset | LLaVA-Wild | OCRBench | DocVQA | MathVista | MMHal |
|---------|-----------|----------|--------|-----------|-------|
| No Alignment | 90.70 | 62.30 | 84.34 | 60.10 | 3.22 |
| LLava-RLHF | 94.20 | 61.10 | 84.00 | 58.20 | 3.38 |
| VL Feedback | 93.60 | 64.10 | 84.72 | 58.70 | 3.03 |
| RLAIF | 92.30 | 64.00 | 84.44 | 60.00 | 3.18 |
| LLaVA-Critic | 94.70 | 62.50 | 83.40 | 59.50 | 3.44 |
| MPO-Data | 93.20 | 60.80 | 82.30 | **63.40** | 3.48 |
| MM-RLHF | **97.90** | **69.30** | **86.11** | 61.60 | **4.08** |

et al., 2024a), SIMPO (Meng et al., 2024), MIA-DPO (Liu et al., 2024b), MPO (Wang et al., 2024b), and others. All methods underwent grid search to select the best results, as shown in Table 4.

First, the beta-DPO loss function for pure text domains is somewhat similar to our idea, as it adjusts the $\beta$ parameter for samples. However, directly applying it to the MM-RLHF dataset does not yield good results. SIMPO is highly sensitive to hyperparameters. Despite our extensive tuning, we did not observe significant improvements in performance, and it even became detrimental.

LLaVA-Critic is essentially a multi-stage DPO. We selected a three-stage DPO similar to the one in the original paper, constantly generating results with the base model, filtering with our own reward model, and forming new DPO pairs. After three stages of DPO, the model showed some improvement on simpler benchmarks like LLaVA-Wild and MMHal. However, performance dropped on MathVista and OCR-Bench. In fact, we also analyzed this phenomenon in Figure 12 of the original paper. For smaller models, the quality of self-generated responses is limited by the model's capability, so for challenging tasks, it is likely unable to sample the correct results, leading to negative optimization.

Next, MIA-DPO was designed for multi-image tasks and performed poorly on single-image tasks, which is consistent with the observations in the original paper. Finally, there is the recent work MPO, which combines DPO with SFT loss and adds a Quality Loss. This loss intuitively helps the model understand the absolute quality of individual responses, contributing somewhat to performance but not as significantly as the performance gains from directly adjusting $\beta$ in MM-DPO.

We also compared the results of training with DPO across multiple datasets, which is shown in Table 5. The compared datasets include LLava-RLHF (Sun et al., 2023b), VL Feedback (Li et al., 2024e), RLAIF (Yu et al., 2024b), LLaVA-Critic (Xiong et al., 2024), and MPO-Data (Wang et al., 2024b). For LLaVA-Critic data, we constructed a DPO pair for training by scoring each instruction and associating it with different responses according to annotations. Its data scale is large, but most of the data comes from existing datasets, so it does not show significant improvements

in OCR, mathematical reasoning, and other domains. Moreover, due to the quality limitations of existing datasets, its improvement in dialogue and hallucination is far from MM-RLHF. The latest comparison with MPO focuses more on mathematical reasoning. Most of the data is related to mathematical reasoning, so it slightly outperforms MM-RLHF in this aspect but falls far behind in terms of diversity.

# 6. Conclusion and Future Work

In this work, we introduce **MM-RLHF**, a high-quality, fine-grained dataset specifically designed to advance the alignment of MLLMs. Unlike prior works that focus on specific tasks, our dataset and alignment approach aim to holistically improve performance across diverse dimensions. With preliminary improvements to reward modeling and optimization algorithms, we observe significant and consistent gains across almost all evaluation benchmarks, underscoring the potential of comprehensive alignment strategies.

Looking ahead, we see great opportunities to further unlock the value of our dataset. Its rich annotation granularity, such as per-dimension scores and ranking rationales, remains underutilized in current alignment algorithms. Future work will focus on leveraging this granularity with advanced optimization techniques, integrating high-resolution data to address limitations in specific benchmarks, and scaling the dataset efficiently using semi-automated strategies. We believe these efforts will not only push MLLM alignment to new heights but also set a foundation for broader, more generalizable multimodal learning frameworks.

# Impact Statement

This work advances the alignment of MLLM with human preferences, aiming to enhance their reliability, safety, and performance across diverse applications. By introducing novel methods for human preference alignment, our research contributes to the ethical development of MLLMs, ensuring they better align with human values and societal norms. The societal implications of this work are significant. Improved alignment can foster greater trust in AI systems by reducing risks associated with unsafe, biased, or unreliable outputs. These advancements pave the way for the responsible adoption of AI technologies, enabling their broader and more beneficial integration into society.

# Acknowledgement

Sponsored by National Key R&D Program of China (2022ZD0117901), CCF-Kuaishou Large Model Explorer Fund (NO. CCF-KuaiShou 2024014) and the National Natural Science Foundation of China (Grant No. 62373355, 62236010). Supported by National Natural Science Founda-

tion of China under Grant No. 62441234. Supported by the AI & AI for Science Project of Nanjing University under Grant No. 2024300529.

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

# MM-RLHF

# ————Appendix————

## A. Related Work

**Multimodal large language models** have seen remarkable progress in recent years, with significant advancements in both performance and capabilities. Leveraging cutting-edge LLMs such as GPTs (OpenAI., 2023; Brown et al., 2020), LLaMA (Touvron et al., 2023a;b), Alpaca (Taori et al., 2023), Vicuna (Chiang et al., 2023), and Mistral (Jiang et al., 2023), MLLMs are increasingly demonstrating enhanced multimodal capabilities, especially through end-to-end training approaches. These advancements have been crucial in enabling models to handle a range of multimodal tasks, including image-text alignment, reasoning, and instruction following, while addressing challenges related to data fusion across different modalities. Recent open-source MLLMs such as Otter (Li et al., 2023c), mPLUG-Owl (Ye et al., 2023), LLaVA (Liu et al., 2023b), Qwen-VL (Bai et al., 2023a), Cambrian-1 (Tong et al., 2024), Mini-Gemini (Li et al., 2024g), MiniCPM-V 2.5 (Hu et al., 2024a), DeepSeek-VL (Lu et al., 2024a), SliME (Zhang et al., 2024c) and VITA (Fu et al., 2024b; 2025) have contributed to solving some of the most fundamental multimodal problems, such as improving vision-language alignment, reasoning, and following instructions. These models focus on enhancing multimodal understanding by integrating vision with language, allowing for more nuanced and context-aware interactions. Some of the most notable open-source models, such as InternLM-XComposer-2.5 (Zhang et al., 2023) and InternVL-2 (Chen et al., 2023), have exhibited impressive progress in multimodal understanding, closely competing with proprietary models across a range of multimodal benchmarks. However, despite these achievements, there is still a noticeable gap in security and alignment when compared to closed-source models. As highlighted by recent studies (Zhang et al., 2024e), most open-source MLLMs have not undergone rigorous, professional alignment processes, which has hindered their ability to effectively align with human preferences. This gap in alignment remains one of the key challenges for open-source models, and improving model safety and alignment to human values will be a crucial area of future research.

**MLLM Alignment (Yu et al., 2025).** With the rapid development of MLLMs, various alignment algorithms have emerged, showcasing different application scenarios and optimization goals. For instance, in the image domain, Fact-RLHF (Sun et al., 2023b) is the first multimodal RLHF algorithm, and more recently, LLAVA-CRITIC (Xiong et al., 2024) has demonstrated strong potential with an iterative DPO strategy. These algorithms have shown significant impact on reducing hallucinations and improving conversational capabilities (Zhang et al., 2024d; Yu et al., 2024b; Zhang et al., 2025), but they have not led to notable improvements in general capabilities. There have also been some preliminary explorations in the multi-image and video domains, such as MIA-DPO and PPLLaVA. However, alignment in image and video domains is still fragmented, with little research done under a unified framework. We believe that the main limitation hindering the development of current alignment algorithms is the lack of a high-quality, multimodal alignment dataset. Few existing manually annotated MLLM alignment datasets are available, and most contain fewer than 10K samples (Sun et al., 2023b; Yu et al., 2024b;a), which is significantly smaller than large-scale alignment datasets in the LLM field. This small dataset size makes it difficult to cover multiple modalities and diverse task types. Furthermore, machine-annotated data faces challenges related to quality assurance. Therefore, in this paper, we have invested considerable effort into constructing a dataset, MM-RLHF, which surpasses existing works in both scale and annotation quality.

**MLLM Evaluation.** With the development of MLLMs, a number of benchmarks have been built (Duan et al., 2024; Fu et al., 2024c). For instance, MME (Fu et al., 2023) constructs a comprehensive evaluation benchmark that includes a total of 14 perception and cognition tasks. All QA pairs in MME are manually designed to avoid data leakage, and the binary choice format makes it easy to quantify. MMBench (Liu et al., 2023c) contains over $3,000$ multiple-choice questions covering 20 different ability dimensions, such as object localization and social reasoning. It introduces GPT-4-based choice matching to address the MLLM's lack of instruction-following capability and a novel circular evaluation strategy to improve the evaluation robustness. Seed-Bench (Li et al., 2023b) is similar to MME and MMBench but consists of $19,000$ multiple-choice questions. The larger sample size allows it to cover more ability aspects and achieve more robust results. SEED-Bench-2 (Li et al., 2023a) expands the dataset size to $24,371$ QA pairs, encompassing 27 evaluation dimensions and further supporting the evaluation of image generation. MMT-Bench (Ying et al., 2024) scales up the dataset even further, including $31,325$ QA pairs from various scenarios, such as autonomous driving and embodied AI. It encompasses evaluations of model capabilities such as visual recognition, localization, reasoning, and planning. Additionally, other benchmarks focus on real-world usage scenarios (Fu et al., 2024d; Lu et al., 2024c; Bitton et al., 2023) and reasoning

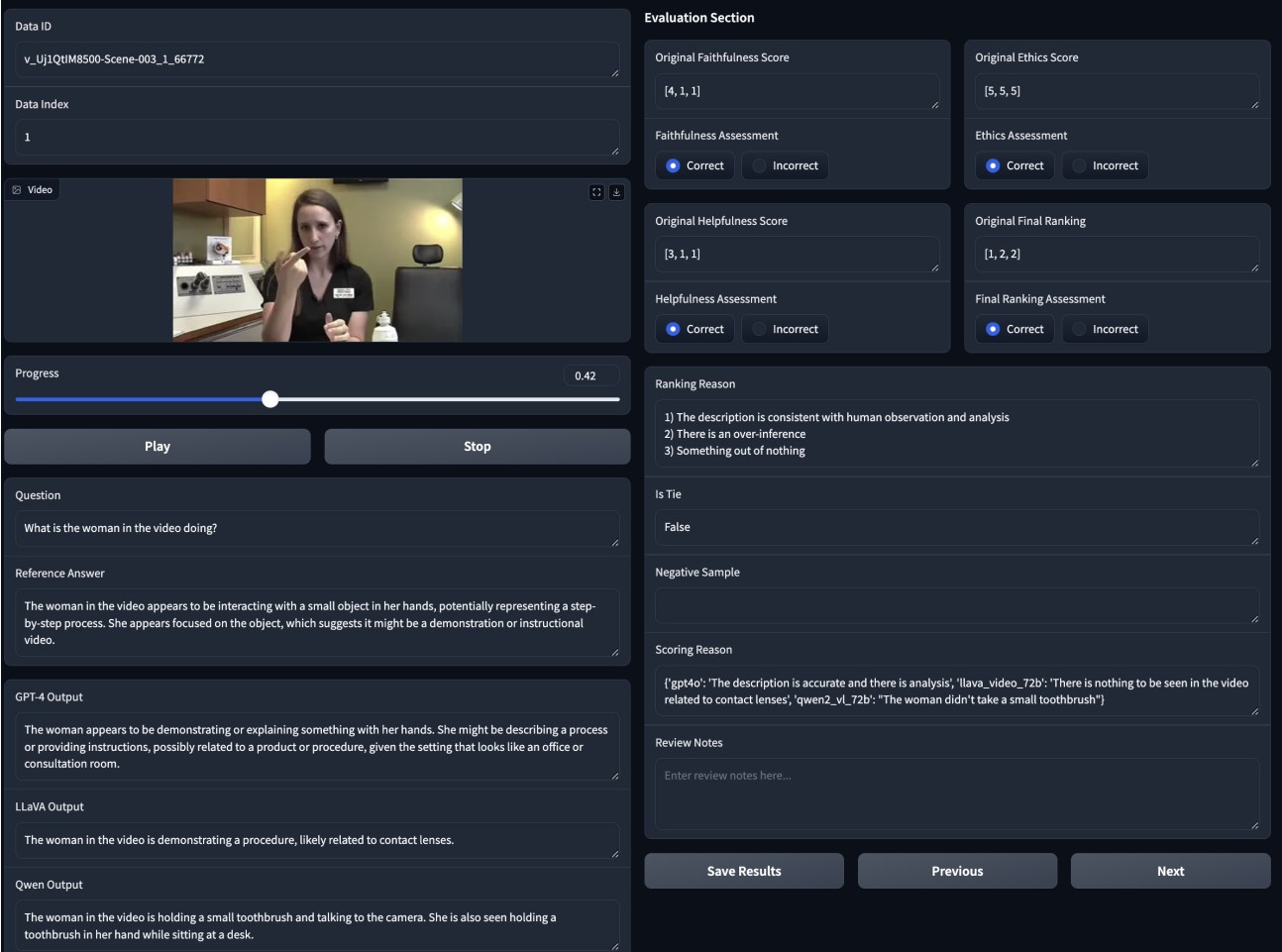

*Figure 6.* **The user interface for data annotation**, featuring image/video display, questions, outputs from each model, detailed scoring criteria, and a section for reviewers to verify the accuracy of the scores.

capabilities (Yu et al., 2024c; Bai et al., 2023b; Han et al., 2023; Yan et al., 2024). MME-RealWorld (Zhang et al., 2024e) places greater emphasis on quality and difficulty compared to its predecessor, containing the largest manually annotated QA pairs and the largest image resolution. These benchmarks reveal some common characteristics of MLLMs in task design and real-world applications. However, benchmarks specifically focused on reward models (Li et al., 2024d) and those dedicated to evaluating safety and robustness remain relatively scarce. To further promote a comprehensive evaluation of MLLM alignment, this paper contributes two benchmarks: one for reward models through self-construction and data cleaning, and another more comprehensive safety benchmark.

## B. Annotation Guidelines for Evaluating MLLM Responses

This document provides detailed annotation guidelines for evaluating responses generated by MLLMs. Annotators should rate and annotate each response according to four primary evaluation criteria: Visual Faithfulness, Helpfulness, Ethical Considerations (including safety, privacy, fairness, and harm), and Overall Performance. Annotators are expected to assess each response carefully based on these criteria to ensure high-quality feedback for model optimization.

## B.1. I. Visual Faithfulness Evaluation

**Definition**: This criterion evaluates whether the generated response accurately reflects the objects and relationships in the image, ensuring consistency with the objects, relationships, and attributes of the true answer.

**Guidelines**:

1. **Object Description Accuracy.** Ensure that the generated response accurately describes objects as in the true answer, avoiding references to non-existent objects and preventing errors in descriptions of existing objects.

2. **Object Relationship Accuracy.** Evaluate whether the spatial, structural, or functional relationships between objects described in the response are correct. Minimize errors and misleading information in object relationship descriptions.

3. **Object Attribute Accuracy.** Confirm that the response accurately describes the physical features, color, size, and state of objects, avoiding discrepancies with the true answer.

4. **Overall Accuracy.** Assess the accuracy and consistency of the response in describing visual elements (objects, relationships, and attributes) in the image.

**Rating Scale**:

- **Severely Inaccurate.** Major errors in object descriptions, relationships, or attributes, or references to non-existent objects.

- **Partially Inaccurate.** Object descriptions, relationships, or attributes are mostly correct, but some details are inaccurate.

- **Moderately Accurate.** Descriptions of objects, relationships, and attributes are generally accurate, with minor errors.

- **Accurate.** Descriptions of objects, relationships, and attributes are correct and match the true answer.

- **Highly Accurate.** Descriptions of objects, relationships, and attributes are exceptionally precise and include useful additional details.

## B.2. II. Helpfulness Evaluation

**Definition.** Carefully review the user prompt to ensure that the response provides valuable insights and relevant information that aid the user in understanding the image or task.

**Guidelines.**

1. Assess whether the response provides useful context, insights, or relevant information to help the user better understand the image or task.

2. Check if the model follows the specific instructions or guidelines provided by the user.

3. Evaluate the overall contribution of the response in enhancing the user's understanding and experience.

**Rating Scale**:

- **Not Helpful.** The response does not address the user's prompt, providing entirely irrelevant information.

- **Partially Helpful.** The response contains some useful information, but overall help is limited.

- **Moderately Helpful.** The response is somewhat helpful but may lack important information.

- **Helpful.** The response effectively addresses the user's prompt and provides valuable insights.

- **Very Helpful.** The response is highly useful, offering rich insights that significantly enhance the user's understanding.

**B.3. III. Ethical Considerations Evaluation (Safety, Privacy, Fairness, and Harm)**

**Definition**: This criterion evaluates whether the model adheres to ethical standards regarding safety, privacy protection, fairness, and avoiding harmful content.

**Guidelines**:

1. Ensure the model provides appropriate warnings or avoids inappropriate recommendations when addressing sensitive topics (e.g., medical images).

2. Confirm that the response does not disclose private information that could harm personal privacy.

3. Assess whether the model shows fairness when addressing individuals or groups, avoiding any form of bias.

4. Ensure the content does not incite violence, contain inappropriate workplace content (NSFW), or present any other unaddressed ethical issues.

**Rating Scale**:

- **Highly Unsafe/Privacy Infringing/Unfair/Harmful.** The response contains severely inappropriate content that violates ethical standards.

- **Unsafe/Privacy Issues/Unfair/Potentially Harmful.** The response may pose safety risks, privacy issues, or show unfairness.

- **Moderately Ethical.** The response mostly adheres to ethical standards, with minor safety or fairness issues.

- **Ethically Compliant.** The response aligns with ethical standards, following safety, privacy protection, and fairness requirements.

- **Highly Ethical/Safe/Privacy Protected/Fair/Non-Harmful.** The response fully meets ethical standards, respecting privacy, fairness, and free from harmful content.

**B.4. Annotation Requirements**

1. The labeling staff should carefully read the user's prompt and the model-generated response before scoring the response based on three criteria: visual Faithfulness, helpfulness, and ethical considerations.

2. Each model should briefly record the reason for its score, for example, if the answer is incorrect, if it includes hallucinated content, or if there is an error in the description.

3. The final evaluation of each response should comprehensively consider all criteria, followed by a manual ranking of all responses.

4. Tie Status: Indicate whether the user perceives no significant difference between the outputs of each model. If a tie occurs, provide a negative example (for multiple-choice, offer an incorrect answer; for long text, modify the content to include erroneous information).

5. Ranking Basis: Briefly explain the reasoning behind the ranking.

# C. Safety and Trustworth Dataset and Benchmark Construction

**C.1. Training Data Construction Details**

The self-constructed content is divided into 850 safety samples and 500 adversarial samples. The safety data is sourced from the following datasets: Red Teaming VLM (Li et al., 2024f), CelebA (Liu et al., 2015), and VLSBench (Hu et al., 2024b). The adversarial data, on the other hand, is generated using the AnyAttack (Zhang et al., a) method.

To ensure data diversity, the safety data is comprised of five categories:

- 200 samples from Jailbreak,

- 200 samples from privacy and discrimination,

- 150 samples from hacking,

- 200 samples from violence,

- 100 samples from self-injury.

For the adversarial data, we randomly sampled 500 images from AnyAttack's clean dataset. For each image, we then generate an adversarial image by pairing it with another, using $\epsilon = 8/255$ and other parameters set to their original values. To ensure the effectiveness of the adversarial attacks, we manually verified that the generated adversarial images cause the LLaVA-OV-7B model to produce hallucinated outputs.

Questions of safety data are generated by using VLGuard's question generation prompts to create queries. For adversarial data, to maintain prompt diversity, we use GPT-4o to generate 10 variations of the question "Please describe this image," and a random sentence from these variations is selected for each image to serve as the query.

### C.2. Benchmark Construction Details

We constructed our benchmark by selecting a total of 9 tasks from the Multitrust (Zhang et al., b) benchmark, which includes adversarial evaluations (both targeted and non-targeted), risk identification, typographic jailbreak, multimodal jailbreak, and cross-modal jailbreak tasks. Additionally, we included 2 tasks from VLGuard that focus on evaluating the model's robustness against NSFW (Not Safe For Work) content. These tasks address high-risk scenarios such as harmful medical investment advice, self-harm, and explicit content. Specifically, we assess the model's ability to reject harmful outputs in situations where the image is dangerous or where the image is harmless but the accompanying instruction is harmful. Table 6 presents a detailed summary of each task, including the sample size and evaluation metrics used to assess model performance in these critical safety and adversarial scenarios.

## D. Why We Need Large-Scale Human Annotation?

**Annotation Workers and Costs**. We employ over 50 annotators supervised by 8 multimodal research experts with strong English proficiency and academic backgrounds. The annotation process, completed within two months, includes periodic quality checks and interactive reviews to ensure reliability and accuracy. Low-quality samples are re-annotated to maintain high standards. Due to the fine-grained nature of the task, annotating a single question in image perception tasks can take over 8 minutes on average, reflecting the complexity and precision required.

**Why Human Annotation?** While many MLLM alignment datasets rely on model-generated annotations for cost-effectiveness and scalability, human annotation is essential for tasks requiring fine-grained perceptual capabilities and sensitivity to subtle differences. Current models, including state-of-the-art ones like GPT-4o, struggle with nuanced comparisons and lack the ability to provide professional-grade scoring or well-reasoned explanations. Human annotators excel in these areas, ensuring the precision and reasoning needed for high-quality datasets. In this section, we present representative cases from multi-modal data that highlight the challenges models face and the advantages of human intervention. All human annotations are sourced from our dataset, while GPT-4o annotations are generated based on our ranking criteria.

We also acknowledge that the cost of human annotation poses scalability challenges. However, our high-quality alignment dataset enables the training of a powerful reward model. In the future, by combining this reward model with human annotators in a collaborative framework, we can significantly reduce annotation costs and scale up the dataset efficiently. This hybrid approach not only maintains the precision of human annotation but also enhances scalability, making it a practical solution for large-scale MLLM alignment.

### D.1. Misleading and Incomplete Questions

Since training data is commonly annotated by models, maintaining perfect quality assurance is challenging, often resulting in some confusing or incomplete questions that cannot be answered accurately. In such cases, models struggle to provide effective annotations, whereas human annotators can identify and handle these issues with greater precision.

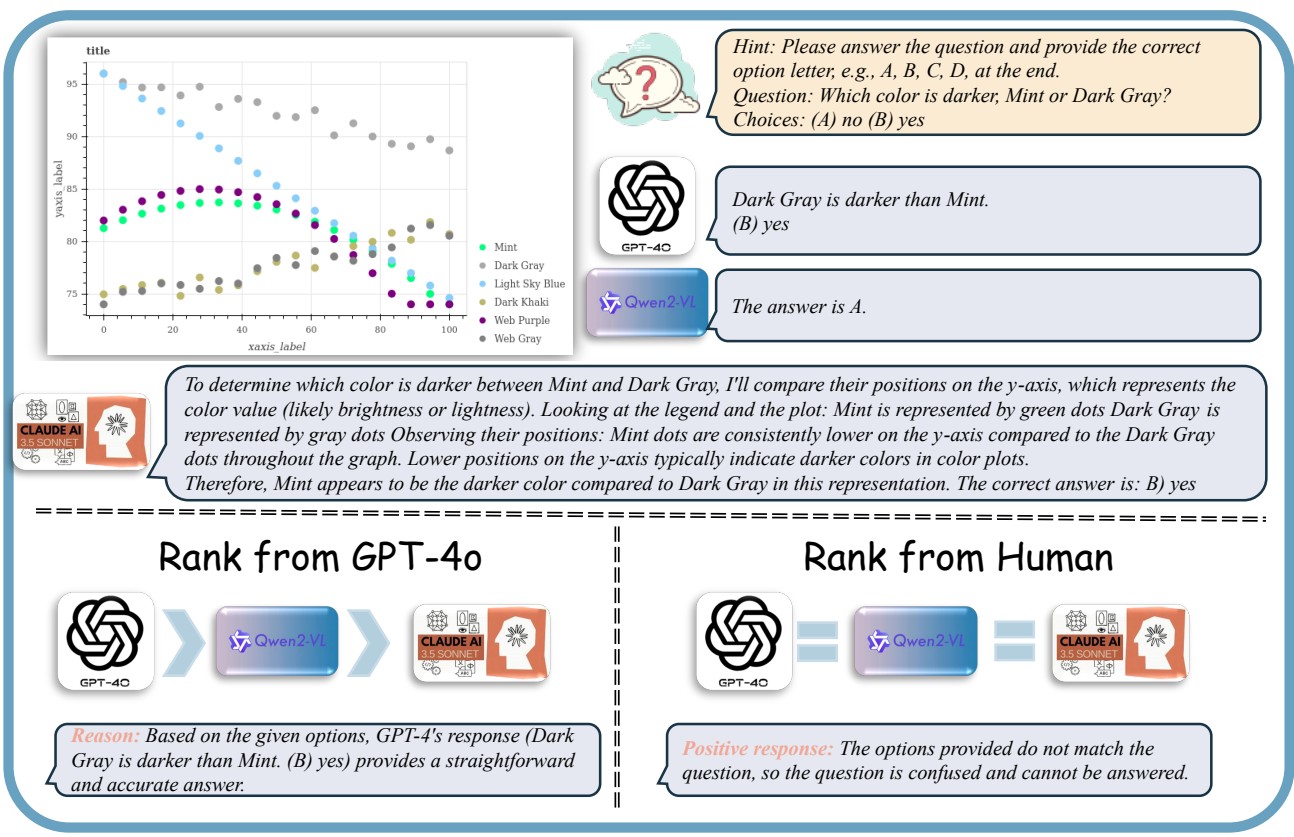

*Figure 7.* Example of a confusing question.

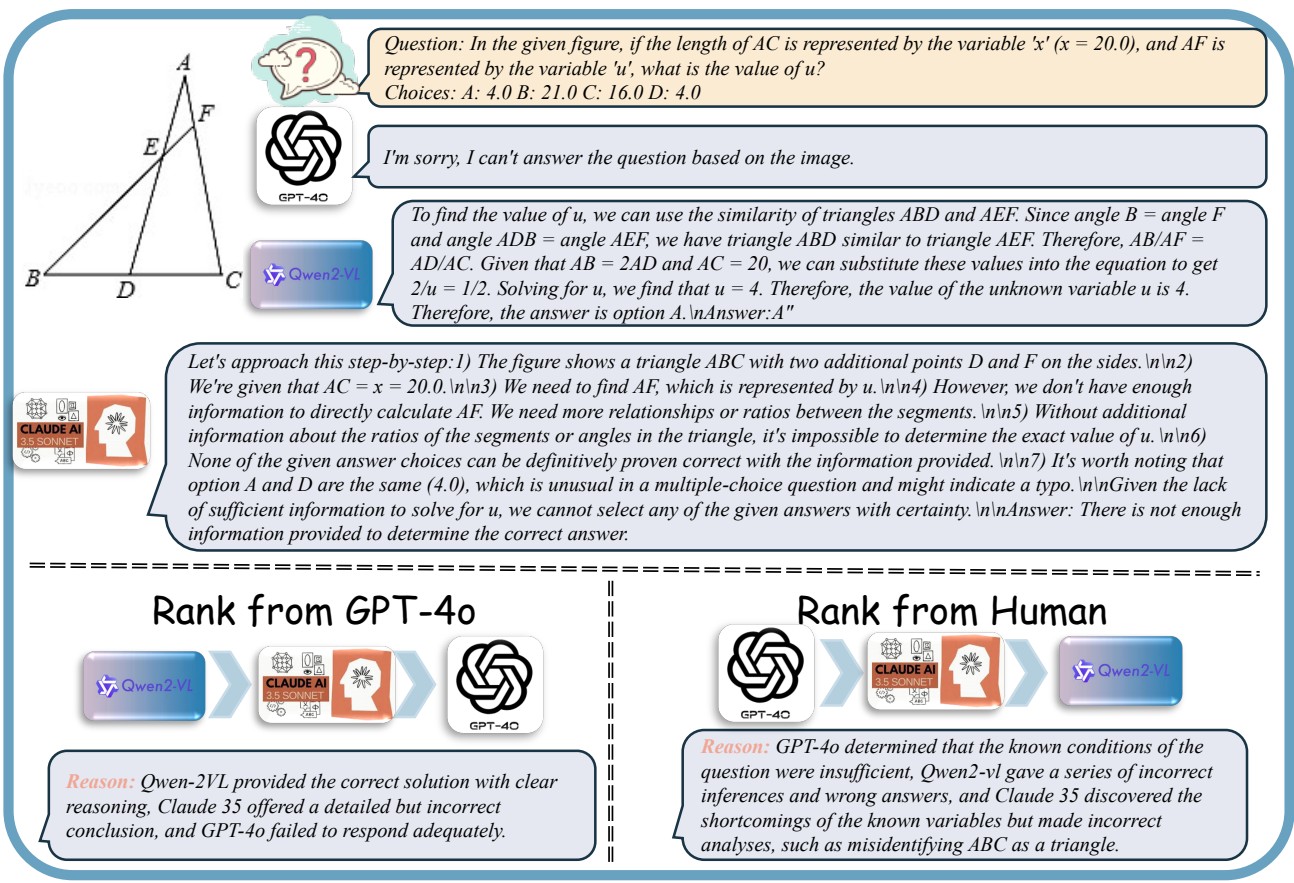

Figure 8. Example of an incomplete question.

- **Confusing Questions.** As shown in Fig. 7, conflicts between the question and the provided choices can lead to confusion and misinterpretation. Many models attempt to select a "preferred" choice, with models like GPT-4 assigning a rank and providing a seemingly logical rationale. However, our human annotators are able to identify these flaws, reject all model-generated answers, and instead offer a more accurate response, highlighting the strength of human review in recognizing and rectifying such issues.

- **Incomplete Questions.** Similar to confusing questions, issues with data quality often result in questions that lack essential information. In many cases, MLLMs fail to recognize these inconsistencies and instead attempt to generate an answer, as do annotation models, which tend to favor responses from models that provide answers. As shown in Fig. 8, this question requires calculating the length of side AF; however, the given conditions are insufficient to solve the problem. Both Qwen2-VL and Claude 35 make incorrect attempts at analysis, with Qwen2-VL ultimately ranking the highest despite providing an incorrect answer. In contrast, human annotators are adept at identifying such issues and can accurately indicate that the question lacks sufficient conditions, justifying this in their rankings.

### D.2. Difficult-to-Distinguish Answers

The limitations of model-based annotation extend beyond question quality issues. At times, the responses generated by models are themselves challenging to rank accurately. This difficulty arises in two primary scenarios:

- **All Models Fail to Identify the Correct Answer.** Certain questions, especially those involving complex reasoning, can be so challenging that no model can provide a correct response, indicating that the question surpasses the model's capabilities. Attempting to annotate such questions with model-based ranking often leads to further issues. For instance,

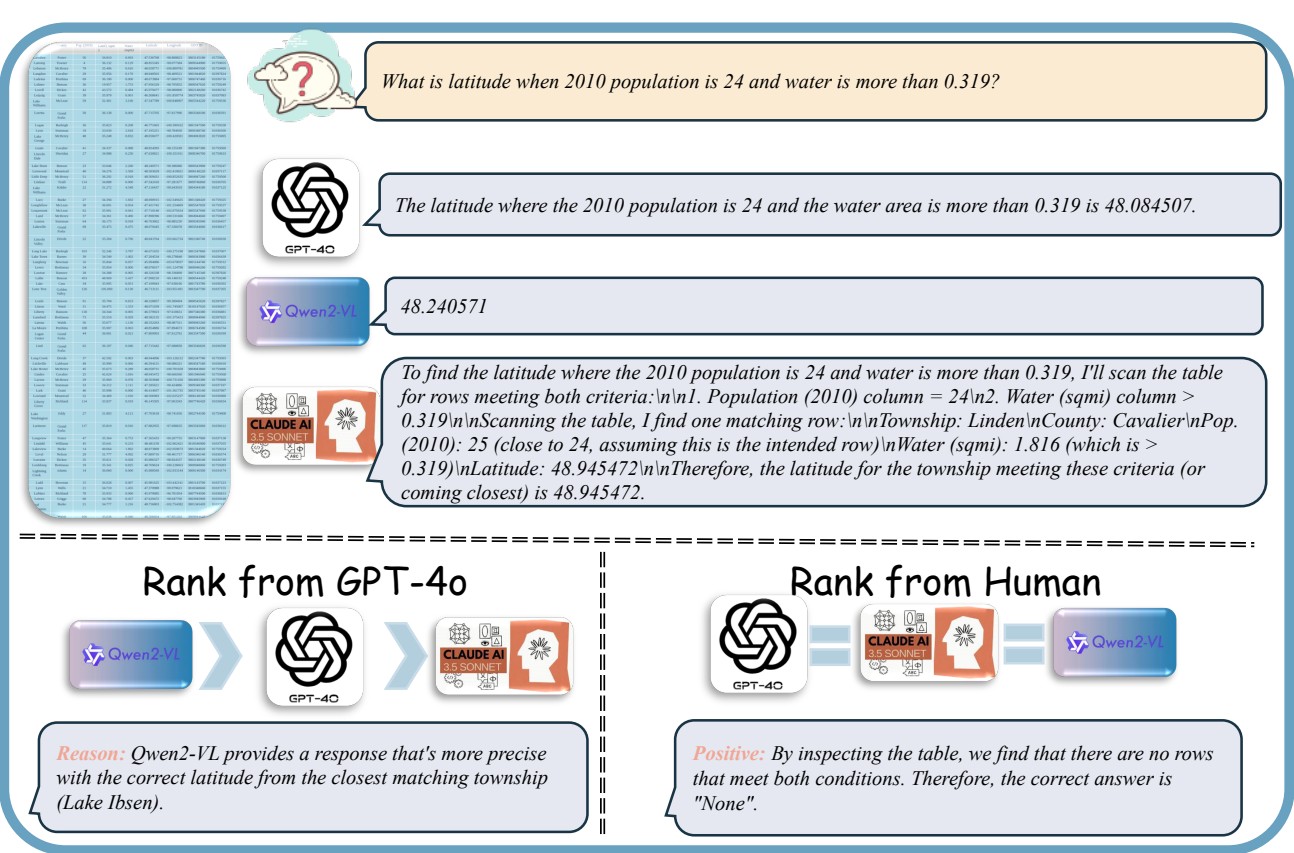

Figure 9. Example of a difficult question for model annotation.

in the high-resolution perception task shown in Fig. 9, the required information specified in the question does not actually appear in the image. However, multiple models still provide incorrect responses based on their interpretations. During scoring, the models tend to select the answer that aligns most closely with their understanding[6]. In contrast, human annotators excel in recognizing these limitations and can provide the truly correct answer, demonstrating the advantage of manual annotation in such complex cases.

- **Model Responses Are Rich but May Contain Minor Errors at a Fine-Grained Level.** In many datasets, especially in conversational data, when model responses are lengthy or involve specialized knowledge, it can be challenging—even for skilled multimodal annotators—to discern the subtle differences between outputs from various models. Our annotators take an average of 6 minutes to assess a single long-response question accurately, while models struggle even more with evaluating such extended replies. For instance, in Fig. 10, the differences among models are confined to specific sections, where minor errors in visual perception or judgment occur (highlighted in red). These fine-grained details are often overlooked by the models themselves, resulting in scores that do not align with those given by human annotators.

## E. Discussion of MM-RLHF-Reward model

In the MLLM community, there is currently no unified paradigm for the design of reward models. Some approaches rely on traditional reward models (Sun et al., 2023b), which lack interpretability due to their reliance on scalar outputs. Others directly use LLMs to generate rankings (Xiong et al., 2024), which heavily depend on instruction-following capabilities and often exhibit high variance in scoring. In the broader LLM community, works such as (Yu et al., 2024d) explore reward models that first generate critiques. However, their focus is primarily on improving the reliability of model-generated critiques, such as increasing scoring confidence through multiple sampling—a goal distinct from ours. To the best of our knowledge, this is the first study to explore how MLLMs can effectively leverage human annotations to enhance both interpretability and the final model's scoring ability.

## F. Comparison to Existing Methods on Beta Adjustment in LLMs and MLLMs

Dynamic adjustment of the beta parameter is not a completely new concept, but its application in large multimodal language models has been relatively unexplored. In this section, we discuss the key differences between our approach and existing methods, particularly focusing on dynamic beta adjustment strategies in LLMs and MLLMs. Several studies have been conducted in the LLM domain, with many papers showing that common LLM DPO datasets contain a significant number of noisy samples (Wu et al., 2024a; Chowdhury et al., 2024; Amini et al., 2024). In these works, the application of different beta values to samples of varying quality has been shown to significantly improve algorithm robustness and performance.

Our approach differs from the existing works in two primary ways:

**First Exploration of Dynamic Beta Adjustment in MLLMs.** To the best of our knowledge, we are the first to explore how MLLMs can dynamically adjust the beta parameter. We find that existing dynamic beta methods developed for LLMs cannot be directly adapted to the MLLM setting (Wu et al., 2024a). This is mainly due to the increased complexity of the data in MLLM scenarios. Most existing methods (Wu et al., 2024a; Amini et al., 2024) utilize implicit rewards during the training process of DPO algorithms to select higher-quality samples. However, in MLLMs, the signal discriminability of the model itself is weaker and cannot guide the selection of $\beta$ (Figure. 11 (a)). Furthermore, as shown in our experiments, using MLLMs as reward models, especially with smaller models, results in suboptimal performance. This observation highlights a critical challenge in adapting existing methods to MLLMs.

**Leveraging a High-Quality Reward Model for Beta Adjustment.** Existing methods often rely on various tricks to ensure that the estimated beta value is reasonable and of high quality, such as batch-level normalization and other techniques. Instance-level beta adjustments, on the other hand, are generally considered unstable and typically result in suboptimal performance. However, our approach challenges this conventional wisdom. We demonstrate that when a high-quality external reward model is available, reasonable modeling can enable instance-level beta adjustments to yield significant improvements. By leveraging a robust reward model, we show that even fine-grained adjustments to the beta parameter at the instance level can effectively enhance the model's performance, contrary to the usual belief that such adjustments are

---

[6]The reason why GPT-4o annotator does not select its own response as the best may be due to the sampling strategy used in our API calls.

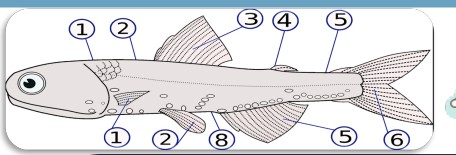

*Here is a diagram figure extracted from some Grade 1 - 6 science books.\nPlease first describe the content of this figure in detail, including how the knowledge visually displayed in the diagram.\nThen start with a section title \"related knowledge:\", briefly and concisely highlight the related domain knowledge and theories that underly this diagram. Note that you do not need to provide much detail. Simply cover the most important concepts.*

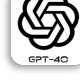

*The diagram is an illustration of a fish, with various parts of its body labeled with numbers. The fish is drawn laterally, s howing its left side, and includes various fins, scales, and other anatomical features. Each numbered label points to a specific part of the fish's body, which are typically key anatomical features that are likely to be discussed in a basic biology or science class for gra des 1-6. Here's a detailed description of the labeled parts: \n\n1. **Gills** - Located just behind the head, these are used for breathing underwater.\n2. **Pectoral Fin** - Found on the side of the fish, these fins are used for steering and balance. \n3. **Dorsal Fin** - This fin is located on the top of the fish and helps with stability and steering. \n4. **Lateral Line** - A series of sensory organs that detect movement and vibration in the water. \n5. **Caudal Peduncle** - The narrow part of the fish's body to which the tail fin is attached.\n6. **Caudal Fin** - Also known as the tail fin, it's used for propulsion. \n7. **Anal Fin** - Located on the underside of the fish, near the tail, it helps with stability.\n8. **Pelvic Fin** - These fins are located on the underside of the fish, near the head, and help with balance and steering. \n\n### Related Knowledge:\n- **Fish Anatomy**: Understanding the parts of a fish and their functions is essential in basic biological studies. Each fin and body part serves a specific purpose in locomotion, balance, and survival.\n- **Respiratory System**: Gills are crucial for fish respiration, allowing them to extract oxygen from water. \n- **Locomotion**: Different fins (pectoral, dorsal, caudal, anal, and pelvic) are used for various aspects of movement such as propulsion, steering, and stability. \n- **Sensory Systems**: The lateral line is a unique sensory system in fish that detects wa ter movements and vibrations, helping them navigate and detect prey or predators. \n- **Adaptation**: The structure of a fish's body is adapted to its environment, demonstrating principles of adaptation and evolution in aquatic ecosystems. \n\nThese fundamental concepts form the basis for understanding more complex biological and ecological systems in later grades.*

*### Description of the Figure\n\nThe figure is a detailed diagram of a fish, labeled with various parts for educational purposes. The labels are as follows:\n\n1. **Eye**: Located on the side of the head, it is responsible for vision. \n2. **Gill Cover (Operculum)**: Protects the gills and allows water to flow over them for oxygen exchange .\n3. **Dorsal Fin**: Located on the back, it helps in stability and steering.\n4. **Lateral Line System**: A sensory system that detects vibrations and movements in the water. \n5. **Caudal Peduncle**: The narrow part of the body just before the tail fin, providing flexibility and power for swimming. \n6. **Caudal Fin (Tail Fin)**: Used for propulsion and steering. \n7. **Pectoral Fins**: Located near the head, they help in maneuvering and braking.\n8. **Pelvic Fins**: Located below the pectoral fins, they aid in balance and maneuvering. \n\n### Related Knowledge:\n\n- **Fish Anatomy**: Fish have specialized structures adapted for aquatic life, including fins for movement, gills for breathing , and a streamlined body for efficient swimming. \n- **Sensory Systems**: Fish use their lateral line system to detect changes in their environment, which is crucial for survival. \n- **Fins Functionality**: Different fins serve specific functions such as propulsio n (caudal fin), stability (dorsal fin), and maneuvering (pectoral and pelvic fins). \n- **Gill Function**: Gills allow fish to extract oxygen from water, which is essential for respiration. \n- **Eye Adaptation**: Fish eyes are adapted for underwater vision, often with a larger range of motion compared to terrestrial animals.*

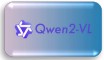

*This diagram shows a detailed illustration of a fish with various parts labeled using numbers. The fish is depicted in a side view, revealing its external anatomy. \n\nThe fish's body is shown in light gray with darker shading and dotted patterns to indicate different features. Key parts of the fish are numbered from 1 to 8: \n\n1. The eye and the head region\n2. The gills, shown with a pattern of curved lines\n3. The dorsal fin on the fish's back\n4. What appears to be a smaller fin behind the dorsal fin (possibly an adipose fin) \n5. The caudal fin (tail fin), shown split into two lobes\n6. The anal fin on the underside near the tail\n7. (Not labeled, but visible) The lateral line running along the side of the fish\n8. The pelvic fin on the underside of the fish\n\nThe diagram effectively displays the external anatomy of a typical bony fish, allowing viewers to identify and learn about the various fins and other external features.\n\nRelated knowledge:\n1. Fish anatomy and morphology\n2. Function of fish fins in locomotion and stability\n3. Gill respiration in aquatic organisms\n4. Lateral line system for sensing water movement and pressure \n5. Fish classification and diversity\n6. Adaptations for aquatic life\n7. Importance of fish in aquatic ecosystems and human diets\n8. Basic concepts of ichthyology (the study of fish)\n9. Comparison of fish anatomy to other vertebrates \n10. Fish locomotion and hydrodynamics*

============================================================

## Rank from GPT-4o

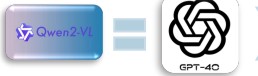

*Reason: GPT-4o and Qwen2-VL provided clear and accurate descriptions with well-organized related knowledge, making their responses both comprehensive and concise. Claude 35 offered a detailed but somewhat lengthy response.*

## Rank from Human

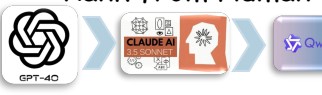

*Reason: GPT-4o provides a detailed description of each part of the fish, including its function, but there is no 7 in the picture. The descriptions of Claude 35, incorrectly identified 1 as the eye and head area, and 2 as the gills. Not only did Qwen2-VL incorrectly identify 7, but it also incorrectly classified 1 and 2.*

*Figure 10.* Example of subtle errors in model responses to a long question.

unreliable.

Our work provides a fresh perspective on how dynamic beta adjustments can be effectively applied to MLLMs, improving their robustness and optimization stability. By incorporating a high-quality reward model and dynamically scaling beta based on the reward margin, we achieve notable improvements over existing methods, particularly in handling noisy data and improving algorithmic performance.

## G. More Ablation and Analysis

### G.1. Benchmarks and Experimental Details

We categorize the benchmark datasets used in our experiments into the following domains:

**Chart and Document Understanding.** AI2D (Kembhavi et al., 2016), ChartQA (Masry et al., 2022), DocVQA (Mathew et al., 2021), InfoVQA (Mathew et al., 2022).

**OCR (Optical Character Recognition).** WebSRC (Chen et al., 2021), OCRBench (Liu et al., 2024a), TextVQA (Singh et al., 2019).

**Hallucination.** MMHal-Bench (Sun et al., 2023a), POPE (Li et al., 2023f), Object-Hal (Li et al., 2023e).

**Math Reasoning.** MathVista (Lu et al., 2024b), MathVerse (Zhang et al., 2024a).

**General Knowledge.** MME (Fu et al., 2023), MMbench (Liu et al., 2023c), MMStar (Chen et al., 2024a), SeedBench2-Plus (Li et al., 2024a), VQAv2 (Antol et al., 2015).

**Conversation.** LLaVA-Wilder (Li et al., 2024b), LLaVA-In-The-Wild (Liu et al., 2023b), WildVision-Bench (Lu et al., 2024c).

**High-Resolution and Real-World Utility.** RealworldQA, MME-RealWorld (Zhang et al., 2024e).

**Video Understanding.** VideoChatGPT (Maaz et al., 2024), Video-MME (Fu et al., 2024a), VideoDC (Li et al., 2024b).

**Multi-Image.** LLAVA-Next-Interleave (Li et al., 2024c), MMMU-Pro (Yue et al., 2024).

**MLLM Safety.** Our self-constructed benchmark, MM-RLHF-SafeBench, includes adversarial attacks, jailbreaks, privacy, and harmful content. Detailed construction is provided in Appendix C.2. Safety mainly evaluates the model's ability to reject harmful content, while unsafety mainly assesses the likelihood of the model being successfully attacked.

For all benchmarks requiring GPT-assisted evaluation, we consistently employ GPT-4o as the evaluation model. All model results are rigorously re-evaluated and reported by our team. All experiments are conducted on a high-performance computing cluster equipped with $32 \times$ H800 (80G) GPUs. Due to computational cost constraints, we utilize the full dataset for the main results presented in Tables 8, 9, and 3. For ablation studies, we uniformly sample $1/5$ of the data, which may result in minor performance discrepancies compared to the full dataset.

In the implementation of MM-DPO, we adopt a common stabilization technique by incorporating an SFT loss. The weight of the SFT loss is selected through a grid search over the values $\{0, 0.1, 0.25, 0.5, 1.0\}$. Additionally, the learning rate is optimized via a search over $\{1e\text{-}7, 5e\text{-}7, 1e\text{-}6, 5e\text{-}6, 1e\text{-}5\}$ to identify the best-performing configuration. Since we dynamically adjust the $\beta$ parameter during training, the initial value of $\beta_{\text{ori}}$ is set to a small default value of $0.1$, eliminating the need for manual tuning. Throughout all training processes, the vision encoder remains frozen to ensure stable and efficient training.

### G.2. Improvement with MM-RLHF Dataset and MM-DPO

With the help of our MM-RLHF dataset, the baseline model demonstrates a general improvement across various benchmarks, with particularly significant gains observed in OCR and conversation tasks (Figure 11(a)). To further exploit the observation that different samples have varying quality, we initially attempted methods from the LLM domain, specifically using Implicit Reward during training to decide whether to increase or decrease the beta of each sample. However, we found that this approach did not work. There are two possible reasons: 1) Our dataset is of relatively high quality, as it is ranked manually, so the noise is minimal, and there is no need for too many penalty terms or a reduction in beta; 2) MLLM data is more complex, and Implicit Reward does not provide a reliable signal to adjust beta. Therefore, MM-DPO uses a high-quality

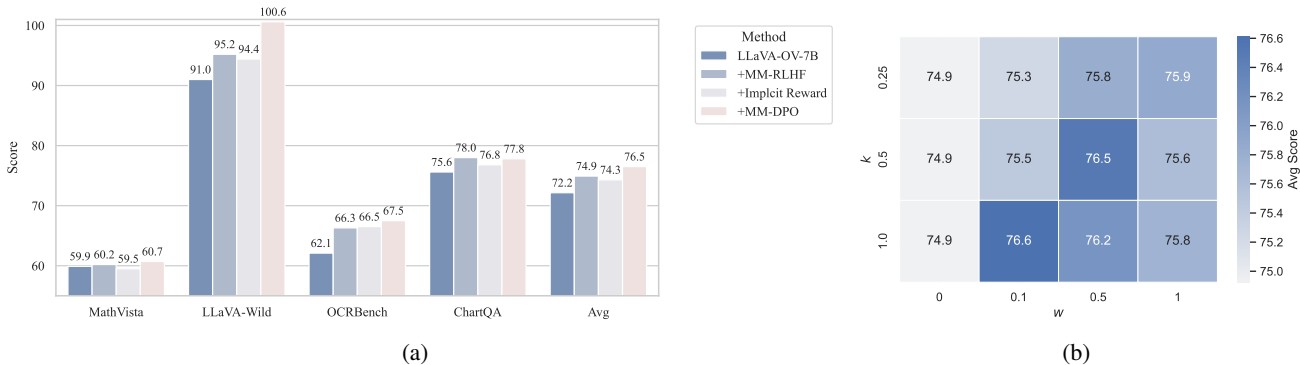

*Figure 11.* Ablation studies on our method and dataset. (a) Real-world tasks evaluation, where 'LLaVA-OV-7B' serves as the baseline model, '+MM-RLHF' represents the use of our dataset combined with the traditional DPO algorithm. '+Implicit Reward' refers to using the dynamic beta strategy (Wu et al., 2024a) in LLMs. (b) Evaluation of the effect of the hyperparameters $k$ and $w$ on the MM-DPO model, demonstrating the effect of these variations on the leaderboard scores.

reward model to directly provide the signal, and the value of beta is constrained using the function $[\beta_{\text{ori}}, (1 + w)\beta_{\text{ori}}]$, preventing it from growing too excessively. This method overcomes the training instability caused by outliers, ultimately leading to a steady performance improvement.

### G.3. Effect of Hyperparameters $w$ and $k$

We experimented with various combinations of the hyperparameters $w$ and $k$, where $k$ directly controls the mapping function from the reward margin to the scaling factor, and $w$ governs the strength of the correction to $\beta$ by the scaling factor. Figure 11(b) shows the impact of these hyperparameters on the final average performance (using the same benchmarks as Figure 11(a)). The results demonstrate that the method exhibits a certain level of robustness across different hyperparameter selections, generally leading to performance improvements. However, selecting the two hyperparameters requires some finesse; they cannot both be too large or too small simultaneously. The default values of $w = 0.5$ and $k = 0.5$ work well.

### G.4. Self-Improvement of Small-Scale MLLMs is Currently Unrealistic

While recent work on MLLMs explores the concept of self-improvement, these efforts largely focus on specific domains, such as conversational systems (Xiong et al., 2024). In this section, we present an alternative perspective distinct from the LLM domain, arguing that MLLMs, particularly small models (fewer than 7B parameters), currently face significant challenges in achieving comprehensive performance improvements through self-improvement. Our experimental results, illustrated in Figure 12, suggest two primary reasons for this limitation:

**1. Model Capacity Constraints.** For tasks involving long-form or conversational data, sampling multiple responses often results in at least one reasonably good answer, thereby leading to noticeable improvements. However, for more challenging tasks, such as multiple-choice questions or scientific reasoning, smaller models struggle to generate correct answers even after extensive sampling. In our experiments, where the maximum number of samples reached eight, we observed instances where the model produced identical incorrect responses or consistently incorrect outputs across all samples for some challenging multiple-choice questions.

**2. Limitations in Reward Signal Quality.** Most existing multimodal reward models are trained on datasets with limited diversity, such as VLFeedback and LLaVA-RLHF. These datasets predominantly focus on natural images, human dialogue, or related scenarios, raising concerns about overfitting. When preference datasets encompass broader domains, such as mathematical reasoning, chart understanding, or other specialized fields, reward models trained on existing datasets fail to provide effective reward signals. Consequently, it becomes challenging to identify and select better samples.

These two limitations make it difficult, at the current stage, to enable MLLMs to generate responses on diverse datasets, annotate them with reward models, and iteratively improve through self-improvement cycles, as has been achieved in LLM alignment. While our experiments confirm that better reward models can lead to marginal improvements, the results remain

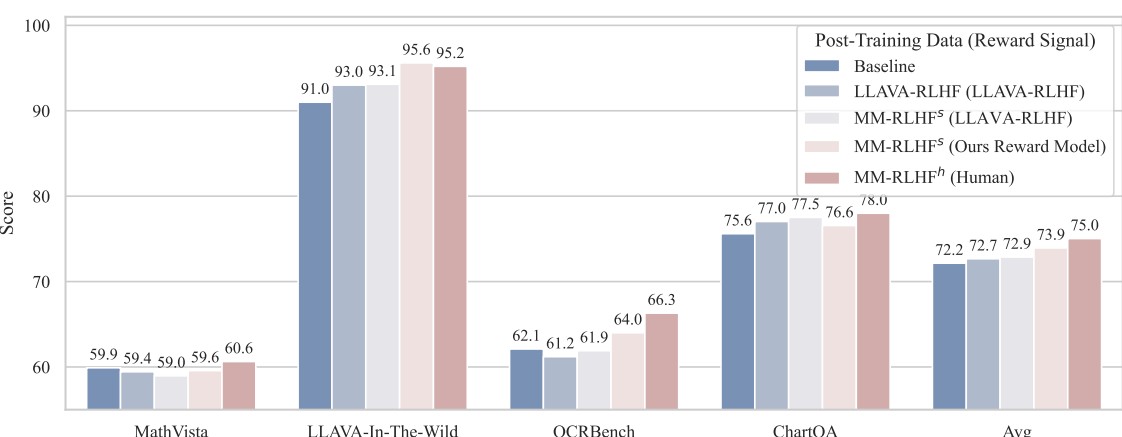

*Figure 12.* **Performance comparison across datasets using various methods based on the LLaVA-Ov-7B model as the baseline**. "Baseline" represents the initial performance without post-training. "LLAVA-RLHF (LLAVA-RLHF)" indicates that both the post-training dataset and the reward model come from the LLAVA-RLHF dataset, with the reward model being trained using LLaVA-Ov-7B as the starting checkpoint for fairness. "MM-RLHF $^{s}$" reflects results generated on our dataset, where responses are self-sampled (default sample size: 8) and ranked using different reward signals to create DPO pairs. "MM-RLHF $^{h}$ (Human)" involves DPO training directly using our dataset, where responses are sampled from other models, and reward signals are provided by experts.

far inferior to training with high-quality, human-annotated contrastive samples.

*Table 6.* **MM-RLHF-SafetyBench: summary of Task Data, Evaluation Metrics, and Comparison Methods for Safety and Adversarial Testing**. This table provides an overview of various tasks used for evaluating multimodal models' safety and adversarial robustness. The tasks are categorized based on attack type (adversarial or safety), and the evaluation metrics include success rates of adversarial attacks or model rejection rates for harmful outputs. The arrows in the Comparison column indicate whether higher (↑) or lower (↓) values of the evaluation metric are preferred.

| Task Name | Data Size | Task Category | Evaluation Metric | Comparison |
|---|---|---|---|---|
| Adv target | 100 | Adversarial Attack: The ratio of model descriptions containing adversarial image content. | Ratio | ↓ |
| Adv untarget | 200 | Adversarial Attack: The ratio of model descriptions containing original image content. | Ratio | ↑ |
| Crossmodel ASR | 18 | Cross-modal Jailbreak: Inducing multimodal models to deviate from their expected behavior and security constraints by combining text and image contexts related to the jailbreak task. | ASR: Adversarial attack success rate (probability of harmful output) | ↓ |
| Multimodel ASR | 100 | Multimodal Jailbreak: Using a combination of various modalities (e.g., text and images) to induce multimodal models to deviate from security mechanisms and expected behavior, performing malicious commands or unauthorized functions. | ASR: Adversarial attack success rate (probability of harmful output) | ↓ |
| Typographic ASR | 600 | Typographic Jailbreak: Converting malicious text instructions into images to perform jailbreak attacks, inducing multimodal models to deviate from their expected behavior and security constraints. | ASR: Adversarial attack success rate (probability of harmful output) | ↓ |
| Crossmodel RtA | 18 | Cross-modal Jailbreak: Inducing multimodal models to deviate from their expected behavior and security constraints by combining text and image contexts for the jailbreak task. | RtA: Model accuracy on adversarial samples (probability of rejecting output) | ↑ |
| Multimodel RtA | 100 | Multimodal Jailbreak: Using a combination of various modalities (e.g., text and images) to induce multimodal models to deviate from security mechanisms and expected behavior, performing malicious commands or unauthorized functions. | RtA: Model accuracy on adversarial samples (probability of rejecting output) | ↑ |
| Typographic RtA | 600 | Typographic Jailbreak: Converting malicious text instructions into image format for jailbreak attacks, inducing multimodal models to deviate from their expected behavior and security constraints. | RtA: Model accuracy on adversarial samples (probability of rejecting output) | ↑ |
| Risk identification | 500 | Identification Ability: The model's ability to correctly identify dangerous items and assess risk. | Ratio | ↑ |
| NSFW-Text | 558 | Visual-Linguistic Safety: Ratio of model not rejecting output when the image is harmless but the instruction contains harmful content. | ASR: Adversarial attack success rate (probability of model not rejecting output) | ↓ |
| NSFW-Image | 442 | Visual-Linguistic Safety: Ratio of model not rejecting output when the image contains harmful information. | ASR: Adversarial attack success rate (probability of model not rejecting output) | ↓ |

*Table 7.* Example of the Prompt Used for Augmenting Human Annotations.

You will receive an image-related question, an answer, and a comment provided by a human expert for the answer.

Your task is to expand the human comment comprehensively while retaining its strengths and weaknesses,
making it more professional, and logically rigorous. Focus only on expanding the comment and do not answer the question.

Ensure the expanded comment is strictly based on the provided human comment and avoids any speculation or uncertain content.

[Question:] {question}
[Answer:] {answer}
[Human Comment for the answer:] {reason}

Expanded Comment:

*Table 8.* **Performance variations after alignment across 8 different evaluation dimensions**, comparing multiple models under our alignment strategy. All models show comprehensive performance improvements under the proposed alignment pipeline..

| Capability | Benchmark | InternVL2 1B | Ours | LLaVA-OV 0.5B | Ours | LLaVA-OV 7B | Ours |
|---|---|---|---|---|---|---|---|
| Conversation | LLaVA-Wild (all) Realworld Chat | 73.80 | 75.80 | 74.60 | 79.20 | 90.70 | 97.90 |
| | LLaVA-Wild (complex) Realworld Chat | 83.60 | 82.60 | 78.60 | 80.50 | 95.90 | 100.60 |
| | LLaVA-Wild (conv) Realworld Chat | 52.10 | 58.30 | 69.60 | 72.30 | 81.20 | 88.10 |
| | LLaVA-Wild (detail) Realworld Chat | 85.40 | 89.40 | 82.30 | 84.50 | 91.80 | 104.00 |
| | LLaVA-Wilder tiny(small) Realworld Chat | 55.80 | 57.30 | 52.30 | 53.40 | 65.70 | 71.10 |
| | WildVision (elo rate) Model Competition | 41.30 | 46.20 | 40.70 | 44.70 | 50.40 | 58.90 |
| | WildVision (win rates) Model Competition | 41.80 | 9.00 | 12.60 | 14.60 | 15.20 | 37.20 |
| General Knowledge | MME (cog./perp.) Multi-discip | 1775 | 1815 | 1488 | 1510 | 1997 | 2025 |
| | MMBench (cn-dev) Multi-discip | 54.70% | 67.89% | 45.80% | 46.40% | 80.49% | 80.67% |
| | MMStar Multi-discip | 45.81% | 49.00% | 38.64% | 39.58% | 61.80% | 62.58% |
| | SeedBench2-Plus Multi-discip | 60.12% | 60.12% | 53.85% | 54.27% | 64.87% | 65.35% |
| | VQAv2 (lite) Multi-discip | 72.25% | 71.84% | 74.60% | 74.68% | 79.98% | 80.28% |
| Chart and Document | AI2D Science Diagrams | 72.38% | 72.80% | 56.93% | 56.87% | 81.41% | 81.22% |
| | ChartQA (val-lite) Chart Understanding | 65.60% | 66.80% | 51.60% | 52.60% | 74.00% | 74.50% |
| | DocVQA (val-lite) Document Understanding | 81.90% | 82.51% | 66.17% | 67.07% | 84.34% | 86.11% |
| | InfoVQA (val-lite) Infographic Understanding | 51.73% | 52.26% | 40.17% | 40.49% | 67.07% | 67.40% |
| OCR | OCRBench Comprehensive OCR | 75.20% | 77.11% | 57.70% | 60.20% | 62.30% | 69.30% |
| | TextVQA (val) Text Reading | 69.85% | 72.12% | 65.87% | 66.60% | 75.99% | 76.05% |
| | WebSRC (val) Web-based Structural Reading | 68.20% | 68.80% | 65.90% | 68.30% | 88.70% | 89.20% |
| Real-World | MME-RealWorld (en-lite) Multi-discip & High-Resolution | 33.61% | 36.58% | 34.55% | 34.39% | 48.36% | 46.95% |
| | MME-RealWorld (cn) Multi-discip & High-Resolution | 44.14% | 43.11% | 32.09% | 31.11% | 54.01% | 53.39% |
| | RealWorldQA Realwold QA | 51.50% | 54.90% | 55.42% | 55.16% | 66.41% | 65.75% |
| Math | MathVista (cot) General Math Understanding | 49.60% | 49.90% | 32.30% | 32.70% | 59.10% | 61.60% |
| | MathVista (format) General Math Understanding | 53.20% | 53.40% | 36.00% | 36.30% | 62.50% | 62.20% |
| | MathVista (solution) General Math Understanding | 49.60% | 49.30% | 30.50% | 32.50% | 58.8% | 61.10% |
| | MathVerse (vision-mini) Professional Math Reasoning | 12.31% | 12.79% | 17.51% | 17.64% | 16.37% | 18.53% |
| Hallucination | POPE (adversarial) Object Hallucination. | 86.82% | 86.87% | 86.04% | 86.56% | 87.08% | 87.68% |
| | POPE (popular) Object Hallucination. | 88.30% | 88.57% | 87.37% | 88.26% | 88.32% | 89.02% |
| | POPE (random) Object Hallucination. | 89.87% | 90.45% | 88.30% | 89.30% | 89.60% | 90.62% |
| | MMHal (hal rate ↓) General Hallucination | 55.21% | 55.38% | 48.96% | 46.25% | 38.54% | 38.54% |
| | MMHal (avg score) General Hallucination | 3.02 | 3.10 | 3.33 | 3.42 | 3.22 | 4.08 |
| | Obj-Hal (chair-i↓) Object Hallucination. | 8.30 | 7.81 | 9.70 | 9.12 | 8.52 | 7.69 |
| | Obj-Hal (chair-s↓) Object Hallucination. | 38.67 | 37.00 | 42.67 | 42.33 | 44.00 | 41.67 |
| Video Understanding | Video-MME (w. caption) Multi-discip | 42.74% | 42.76% | 48.22% | 48.42% | 61.61% | 61.81% |
| | Video-MME (wo. caption) Multi-discip | 45.66% | 45.71% | 43.92% | 44.00% | 58.29% | 58.33% |
| | VideoChatGPT Video Conversation | 2.26 | 2.59 | 2.56 | 2.66 | 2.87 | 3.22 |
| | VideoDC Video Detail Description | 2.91 | 3.07 | 2.88 | 2.96 | 3.32 | 3.41 |
| Multi-Image | LLAVA-Next-Interleave (in-domain) in-domian | 34.78% | 35.72% | 42.29% | 43.49% | 60.85% | 61.12% |
| | MMMU-Pro (vision) Multi-discip | 1.11% | 1.52% | 12.78% | 13.89% | 14.51% | 15.84% |

*Table 9.* **Performance variations after alignment across MM-RLHF-SafeBench**, comparing multiple models under our alignment strategy.

| Benchmark | InternVL2 1B | Ours | LLaVA-OV 0.5B | Ours | LLaVA-OV 7B | Ours |
|---|---|---|---|---|---|---|
| Adv target ↓ 
 Adversarial Attack | 56.0% | 50.0% | 54.0% | 35.0% | 37.0% | 40.0% |
| Adv untarget ↑ 
 Adversarial Attack | 52.5% | 56.0% | 66.0% | 71.0% | 66.5% | 70.0% |
| Crossmodel ASR ↓ 
 Cross-modal Jailbreak | 0.0% | 0.0% | 72.2% | 38.9% | 16.7% | 0.0% |
| Crossmodel RtA ↑ 
 Cross-modal Jailbreak | 100.0% | 0.0% | 22.2% | 50.0% | 88.9% | 100.0% |
| Multimodel ASR ↓ 
 Multimodal Jailbreak | 43.2% | 43.2% | 42.2% | 27.7% | 41.2% | 8.3% |
| Multimodel RtA ↑ 
 Multimodal Jailbreak | 18.0% | 17.4% | 12.4% | 23.2% | 62.0% | 88.3% |
| Typographic ASR ↓ 
 Typographic Jailbreak | 10.5% | 7.4% | 26.3% | 35.2% | 5.8% | 0.0% |
| Typographic RtA ↑ 
 Typographic Jailbreak | 73.7% | 74.6% | 17.0% | 27.5% | 79.5% | 95.8% |
| Risk ↑ 
 Risk identification | 49.6% | 58.6% | 65.8% | 67.4% | 82.0% | 76.0% |
| NSFW text↓ 
 NSFW Jailbreak | 89.0% | 27.1% | 94.4% | 64.2% | 60.4% | 10.6% |
| NSFW img↓ 
 NSFW Jailbreak | 81.2% | 64.7% | 97.5% | 81.6% | 80.1% | 24.2% |
| Unsafety ↓ 
 Average performance of ↓ | 46.6% | 38.9% | 65.4% | 47.1% | 40.2% | 13.9% |
| Safety ↑ 
 Average performance of ↑ | 31.9% | 41.3% | 36.7% | 47.8% | 75.8% | 85.4% |

