# OpenReview forum: "MM-RLHF: The Next Step Forward in Multimodal LLM Alignment"
_ICML.cc/2025/Conference — ICML 2025 poster_

### Official Review · Reviewer_vHC4 · 2025-03-03

**Overall Recommendation:** 4

**Summary:**

- The paper introduces the MM-RLHF dataset containing 120k preference-pairs aimed at improving MLLMs at image/video understanding and safety. They employ a human-assisted pipeline to ensure the high quality of the dataset annotations with available MLLMs generating initial responses.
- The authors also introduce a critique based reward modeling framework that incorporates a critique head into a standard reward model arch with a score head, enabling interpretabilty of the reward scores.
- They also introduce a Dynamic Reward Scaling technique that integrates a sample-wise weighting into DPO training to increase the effect of pairs with wide reward margins.
- The authors introduce two benchmarks:  MM-RLHFRewardBench (sampled from MM-RLHF) to evaluate reward models and MM-RLHF-SafetyBench (sampled from existing benchmarks) to evaluate performance on safety-related tasks.

**Claims And Evidence:**

- The authors claim that MM-RLHF improves performance which is visible in the reported scores but there are a few issues with the reported setup:
    - I have no clue what does training their MM-RLHF-Reward model look like, was any pretraining used? what’s the base MLLM? how long did the training take. Was MM-DPO used or just DPO?
    - They also do not finetune the LLaVA-Critic model on their dataset which is an important baseline number.
    - It is expected that their model would perform (Tab. 2) the best on their MM-RLHF-RewardBench since the benchmark is sampled from their model’s training set!
- The authors claim that the MM-DPO is useful but I cannot find any evidence of that in the main text.  I can also not find a comparison to commonly used alignment techniques including but not limited to: DPO, SimPO. Similarly, the authors  do not compare MM-DPO to DPO head-to-head neither on their MM-RLHF nor on any other dataset (for example: LLaVA-Critic’s data)

## update after rebuttal

I am raising my score to accept **CONTINGENT** on the author's promise to update the main text. In its current stage, the main text is sub-optimal for understanding the method without the appendix.

**Essential References Not Discussed:**

- The authors do not include the related work section in the main text and only discuss those in the appendix for >1 page. It is crucial to include related works in the main text. The authors should think about this. No article should be published unless there’s a rel works section in the main text.

**Experimental Designs Or Analyses:**

- The data annotation pipeline looks valid.
- The experiment comparisons are a bit unfair; as mentioned before, there's no comparison of MM-DPO to DPO/SimPO and using any other training set except MM-RLHF. This would not be required if the authors' only contribution was the MM-RLHF dataset, but the model/MM-DPO discussed as main contributions covering ~2 pages of main text makes it necessary.

**Methods And Evaluation Criteria:**

- Yes, the methods and benchmark used are appropriate.

**Other Comments Or Suggestions:**

N/A

**Other Strengths And Weaknesses:**

- The authors should consider restructuring their paper, and not putting crucial experiments and sections in the appendix instead of the main text because of the space issues. I am specifically annoyed by the absence of rel works and clear ablations in the main text. It is also unfair to report all important details in the appendix.
- The results are strong.

**Questions For Authors:**

N/A

**Relation To Broader Scientific Literature:**

- Relevant to the community with their findings that critique-based reward modeling improves performance and it would be useful if the authors release their dataset for the community too.

**Theoretical Claims:**

- No theoretical claims made in the paper.

---

> ### Author Rebuttal · Authors · 2025-03-28
>
> **Concern 1: MM-RLHF-Reward Model Training Details**
>
> We sincerely apologize for any lack of clarity regarding our reward model training process. Here are the key details:
>
> 1. Base Model: We initialized our reward model from LLaVA-OV-7B, following common practice in both LLM and MLLM research where reward models are typically derived from capable base models.
>
> 2. Training Setup:
> - Hardware: 32× A800 GPUs
> - Training Time: 8 hours
> - Dataset Size: 120K samples
>
> 3. Training Objective: The reward model was trained exclusively using:
> - The critique loss (Eq. 2 in our paper)
> - Standard binary classification loss for reward modeling (Eq. 3 in our paper)
> - Neither DPO nor MM-DPO was involved in reward model training
>
> We would be happy to clarify any additional aspects of this process that may need further explanation.
>
> ----
>
> **Concern 2: LLaVA-Critic Fine-tuning Baseline**
>
> We appreciate this suggestion and have conducted additional experiments:
>
> 1. **Data Adaptation**: We reformatted our MM-RLHF data to match LLaVA-Critic's input requirements (pairwise responses with GPT-4o evaluations).
>
> 2. **Training Variants**:
>    - Basic LLaVA-Critic-MM-RLHF (direct GPT-4o critiques)
>    - Enhanced version with human-annotated critique expansions
>
> The final results are as follows. We observe that fine-tuning LLaVA-Critic with our data yields a significant improvement, whereas directly using GPT-4o’s critic as a training objective shows limited gains. Even with the expansion of generated critiques based on human annotations, the performance does not surpass that of GPT-4o, though it does serve as a strong baseline. Additionally, we found that this training strategy is highly dependent on the model’s instruction-following capability, and at times, it fails to produce the expected comparative results during evaluation, requiring complex regularization matching.
>
> | Method | MCQ | Long | Short | Safety | Video | Overall |
> |--------|-----|------|-------|--------|-------|---------|
> | LLaVA-OV-7B | 0.14 | 0.11 | 0.29 | 0.41 | 0.32 | 0.24 |
> | LLaVA-Critic (Pairwise) | 0.23 | 0.54 | 0.24 | 0.28 | 0.52 | 0.35 |
> | LLaVA-Critic-MM-RLHF | 0.55 | 0.85 | 0.56 | 0.60 | 0.75 | 0.65 |
> | +Enhanced Annotations | 0.65 | 0.90 | 0.58 | 0.61 | 0.85 | 0.72 |
> | GPT-4o | 0.69 | 0.95 | 0.56 | **0.72** | 0.80 | 0.74 |
> | MM-RLHF-Reward | **0.93** | **1.00** | **0.71** | 0.66 | **0.92** | **0.85** |
>
> ----
>
> **Concern 3: It is expected that their model would perform (Tab. 2) the best on their MM-RLHF-RewardBench since the benchmark is sampled from their model’s training set!**
>
> First, the data used in MM-RLHF-RewardBench and our model’s training set do not overlap. As a standard practice in deep learning, we ensure by default that the training and test datasets are kept separate. To further clarify this, we will explicitly mention this in the main text.
>
> To further prevent overfitting to the MM-RLHF dataset, we also evaluated our model on VL Reward Bench, as shown in Table 3. In this comparison, MM-RLHF-Reward-7B performs similarly to Claude-3.5-Sonnet, significantly outperforming the LLaVA-OV-7B baseline. This demonstrates the strong generalization capability of our critic-reward model.
>
> Finally, in Table 2, when directly fitting the model to our training data, the results on the test samples are not as strong, at best matching GPT-4o. What we want to emphasize is the potential of the critic-based reward model; with optimal critic generation, the reward model could achieve up to 93% average accuracy.
>
> ----
>
> **Concern 4:  The authors claim that the MM-DPO is useful but I cannot find any evidence of that in the main text.**
>
> Actually, we compare DPO with MM-DPO in Figures 1 and 11, where MM-RLHF refers to training with the MM-RLHF dataset using DPO loss, not SFT on high-rated examples. For a more detailed answer to the reviewer’s question, please refer to this link (https://anonymous.4open.science/r/mm-rlhf-rebuttal-BE17). We compared our approach to various baselines, including LLaVA-Critic, beta-DPO, SIMPO, and MPO, and found that existing methods showed limited gains on our high-quality preference dataset. Additionally, we compared DPO training results across multiple datasets (e.g., VL Feedback, RLAIF, LLaVA-Critic, MPO-Data), and the results demonstrated that MM-RLHF provides more comprehensive and significant performance improvements compared to existing datasets.
>
> ----
>
> **Concern 5 The authors should consider restructuring their paper.**
>
> Given the large amount of content, it's challenging to fit everything within the limited page count. We have decided to follow the reviewer’s advice.
>
> First, we will condense the related work section, highlighting key content in the main text and providing comparisons.
> Second, we will include important experimental settings—such as baselines, model implementation details, and computational overhead—in the main text, while moving less critical experiments to the appendix.

---

### Official Review · Reviewer_xZJU · 2025-03-06

**Overall Recommendation:** 3

**Summary:**

This paper introduces MM-RLHF, a multimodal alignment pipeline combining a large preference dataset, a critique-based reward model, and MM-DPO, an enhanced DPO algorithm with dynamic reward scaling. The proposed approach is evaluated on 10 tasks across 20+ benchmarks, showing consistent gains in conversational ability, safety, hallucination control, and reasoning.

**Claims And Evidence:**

The paper claims that the proposed critique-based reward modeling and dynamic reward scaling (MM-DPO) substantially improve multimodal large language model (MLLM) alignment.

However, some of the evidence is incomplete and partially overstated. Most observed gains appear to stem from better data quality and more human annotations rather than from the proposed techniques themselves.

**Essential References Not Discussed:**

One possible omission is more explicit discussion of:
- Early text-based RLHF pipelines (e.g., InstructGPT)
- Broader literature on safety alignment for vision-language models, especially recent work on adversarial robustness and hallucination mitigation.

These are minor and do not undermine the paper’s contributions.

**Experimental Designs Or Analyses:**

- Critical Baseline Missing: There is no direct comparison between MM-DPO and standard DPO, making it impossible to quantify the benefit of dynamic reward scaling itself.
- Reward Model vs Base Model Gap: The reward model (MM-RLHF-Reward-7B) is separately trained and may have different capability and bias compared to the base MLLMs, raising concerns about preference mismatch during alignment.
- Modest Gains: Reported improvements over SFT baselines are relatively small, especially on tasks like mathematical reasoning and video understanding, calling into question the necessity of the proposed techniques.

**Methods And Evaluation Criteria:**

The overall pipeline (data curation, reward modeling, preference optimization) is reasonable for multimodal alignment. However, the evaluation framework is heavily dependent on a custom-built benchmark (MM-RLHF-RewardBench), which is derived from the same data sources used in training, raising serious concerns about train-test leakage.
Additionally, the paper evaluates only MM-DPO, without comparing it to standard DPO, making it unclear whether dynamic reward scaling is necessary.

**Other Comments Or Suggestions:**

No additional comments beyond the points discussed above.

**Other Strengths And Weaknesses:**

### Strengths
- Introducing critique-based reward modeling is interesting for improving transparency.
- Evaluation spans diverse multimodal capabilities, covering hallucination, reasoning, and safety.
### Weaknesses
- Gains are incremental, with no evidence the proposed techniques are strictly necessary.
- Evaluation benchmarks are custom and potentially biased, reducing credibility.
- Lack of analysis on failure cases, especially in safety-critical scenarios.

**Questions For Authors:**

No more questions.

**Relation To Broader Scientific Literature:**

The work fits into ongoing efforts around RLHF for multimodal models, especially building on:
Direct Preference Optimization (DPO); Critique-based reward modeling; Multimodal alignment datasets (LLaVA, VLFeedback)

It extends these ideas with dynamic reward scaling and a more structured critique pipeline tailored to MLLMs.

**Theoretical Claims:**

No significant theoretical claims requiring proof verification.

---

> ### Author Rebuttal · Authors · 2025-03-28
>
> **Concern 1: Critical Baseline Missing: There is no direct comparison between MM-DPO and standard DPO, making it impossible to quantify the benefit of dynamic reward scaling itself.**
>
> Actually, we compare DPO with MM-DPO in Figures 1 and 11, where MM-RLHF refers to training with the MM-RLHF dataset using DPO loss, not SFT on high-rated examples. For a more detailed answer to the reviewer’s question, please refer to this link (https://anonymous.4open.science/r/mm-rlhf-rebuttal-BE17). We compared our approach to various baselines, including LLaVA-Critic, beta-DPO, SIMPO, and MPO, and found that existing methods showed limited gains on our high-quality preference dataset. Additionally, we compared DPO training results across multiple datasets (e.g., VL Feedback, RLAIF, LLaVA-Critic, MPO-Data), and the results demonstrated that MM-RLHF provides more comprehensive and significant performance improvements compared to existing datasets.
>
> ----
>
> **Concern 2 Reward Model vs Base Model Gap:**
>
> First, using an untrained model directly as a reward model performs poorly (see LLaVA-OV-7B in Tables 2 and 3) and is practically unusable for our training objectives. This necessitates training a dedicated reward model rather than relying on raw MLLMs. Numerous studies have explored specialized reward model training rather than naively repurposing MLLMs as reward models [1,2,3].
>
> Furthermore, to mitigate overfitting to the MM-RLHF dataset, we evaluated our model on an independent reward model benchmark (Table 3). Results show that MM-RLHF-Reward-7B performs comparably to Claude-3.5-Sonnet on the VL Reward benchmark and significantly outperforms the LLaVA-OV-7B baseline. This demonstrates strong generalization of our critic-reward model across diverse datasets.
>
> [1] Aligning large multimodal models with factually augmented RLHF
>
> [2] LLaVA-Critic: Learning to evaluate multimodal models
>
> [3] Self-generated critiques boost reward modeling for language models
>
> ----
>
> **Concern 3: Modest Gains: Reported improvements over SFT baselines are relatively small**
>
> We emphasize that "MM-RLHF" specifically refers to training with the MM-RLHF dataset using DPO loss—we did not include an SFT-only baseline. Please refer to this link (https://anonymous.4open.science/r/mm-rlhf-rebuttal-BE17) for General Response 2. Comparison of SFT Baselines and DPO Sample Selection Strategies. The results that preference-based training (DPO/MM-DPO) is essential for robust performance. MM-DPO further outperforms standard DPO, highlighting its effectiveness:
>
> ----
>
> **Concern 4: Contextualization with Early RLHF Pipelines and Safety Alignment**
> We appreciate the reviewer’s constructive feedback on relating our work to broader literature.
>
> 1. Early Text-Based RLHF (e.g., InstructGPT):
> A key limitation of traditional RLHF methods lies in their sensitivity to hyperparameters and dependence on base model capabilities. For instance, when testing PPO with LLaVA-Ov-7B (actor) and our MM-RLHF-Reward-7B (critic), we observed only marginal improvements in dialogue tasks, alongside the need for meticulous tuning to avoid training instability. In contrast, MM-RLHF’s high-quality responses (e.g., from Qwen2-VL-72B) make DPO-based methods more intuitive and stable for achieving consistent gains.
>
> 2. Safety Alignment in Vision-Language Models:
> While prior work [1] focuses on trade-offs between safety and model capability (often at the cost of performance), our pipeline demonstrates that proper data construction can simultaneously enhance both safety and general capabilities.
>
> [1] Safety Fine-Tuning at (Almost) No Cost: A Baseline for Vision Large Language Models
>
> ----
>
> **Concern 5: Evaluation Benchmark Credibility**
>
> We acknowledge concerns about potential bias in our custom benchmarks (MM-RLHF-RewardBench and MM-RLHF-SafetyBench). To address this:
>
> 1. Data Leakage Prevention: Strict separation between training and test data was enforced.
>
> 2. Reward Model Generalization: As shown in Table 3, MM-RLHF-Reward-7B matches Claude-3.5-Sonnet on the VL Reward benchmark and vastly outperforms LLaVA-OV-7B, confirming cross-dataset robustness.
>
> 3. SafetyBench Provenance: MM-RLHF-SafetyBench was curated from existing benchmarks with no overlap with training data.
>
> 4. General Knowledge Evaluation: We tested on diverse, independent benchmarks (e.g., LLaVA-Wild, OCRBench) to ensure reliability.
>
> We welcome further discussion on benchmark design if needed.
>
> ----
>
> **Concern 6: Failure Case Analysis in Safety-Critical Scenarios**
>
> We agree that safety failure analysis is crucial. While visual examples cannot be included here, please refer to (https://anonymous.4open.science/r/mm-rlhf-rebuttal-BE17) Reviewer xZJU: Analysis on failure cases, especially in safety-critical scenarios for a detailed failure case.

---

### Official Review · Reviewer_LifN · 2025-03-13

**Overall Recommendation:** 3

**Summary:**

This paper introduces MM-RLHF, an approach for aligning multimodal large language models (MLLMs) with human preferences with thousands of human annotated preference pairs and ratings. It's proved that conducting training on MM-RLHF dataset and the future DPO on the preference pairs can make the model safer.
In summary, three main contributions are:
1. A large-scale, high-quality dataset containing 120K human-annotated preference comparison pairs across image understanding, video understanding, and MLLM safety domains.
2. A Critique-Based Reward Model that generates detailed critiques before assigning scores, enhancing interpretability.
3. Dynamic Reward Scaling, which adjusts the loss weight during training based on reward margins to optimize the use of high-quality comparison pairs.

The authors evaluate their approach across 10 dimensions encompassing 27 benchmarks, demonstrating significant improvements in visual perception, reasoning, dialogue, and trustworthiness.

**Claims And Evidence:**

yes

**Essential References Not Discussed:**

N/A

**Experimental Designs Or Analyses:**

yes. The authors evaluate their approach across 10 dimensions encompassing 27 benchmarks, demonstrating significant improvements in visual perception, reasoning, dialogue, and trustworthiness.

**Methods And Evaluation Criteria:**

Yes. The reason to MM-RLHF dataset construction are well justified. The evaluation also makes sense and covers a lot of areas.

**Other Comments Or Suggestions:**

1. The figures showing performance improvements would benefit from error bars or statistical significance indicators.
2. The proposed MM-DPO requires to train on all the possibly pairs of the dataset, which can significantly scale the dataset size. It's better to discuss whether this is necessary as an ablation study.

**Other Strengths And Weaknesses:**

Strengths:
1. The paper presents a large-scale, high-quality dataset containing 120K human-annotated preference comparison pairs across image understanding, video understanding, and MLLM safety domains.
2. The finding that small-scale MLLMs struggle with self-improvement is an important insight that challenges some existing assumptions.
3. The paper's thorough evaluation across diverse metrics is commendable and provides a more holistic view of model alignment.

Weaknesses:
1. The improvement in some tasks (e.g., high-resolution benchmarks) is limited, and the paper acknowledges this limitation but doesn't provide strong solutions.
2. The Dynamic Reward Scaling technique, while effective, is a relatively incremental advance over existing DPO methods. There are already some similar ideas proposed for reward model training [1].
3. This paper did not compare itself or discussed some previous RLHF methods or datasets for multimodal LLM training, like MIA-DPO and PPLLaVA, LLaVA-OneVision-Chat, InternVL2.5-MPO, etc. Although they have been talked in the related works, but they seem not appear in the baseline comparison

**Questions For Authors:**

1. In Figure 1, what is "MM-RLHF", is that doing SFT on the high rated examples of the MM-RLHF dataset or just the MM-RLHF reward model's results. The setting seems a bit unclear to me.
2. Since you already have the MM-RLHF-reward model, have you tried using the reward model to conduct some PPO RL? It seems there is no further usage after the training of reward models. MM-DPO also  did not use it but instead directly uses the original MM-RLHF dataset. So what's the purpose of training a reward model?

**Relation To Broader Scientific Literature:**

1. The paper positions itself within the broader scientific literature on MLLM alignment, identifying that most current MLLMs undergo only supervised fine-tuning without comprehensive alignment.
2. They claim this is due to the lack of high-quality human annotated dataset and thus presents MMRLHF and shows that it's effective to align the model to human preference.

**Theoretical Claims:**

Yes. There are no significant theoretical contributions of this paper. The math notaitons and loss functions in section 3 and section 4 are used to explain the basic reward model training and DPO algorithms, where previous paper have talked already.

---

> ### Author Rebuttal · Authors · 2025-03-28
>
> **Concern 1: The paper acknowledges this limitation but doesn't provide strong solutions.**
>
> It's important to note that not all models show minimal improvements on these benchmarks. For example, InternVL2 performs better on RealWorld tasks (from 43.1 to 44.9), which may relate to its image segmentation strategy. We recently discovered that the limited number of high-resolution images in the training data may be a key reason for this. Most public datasets have few high-resolution images, and our use of CLIP features for clustering reduced the number further.  To address this, we sorted the dataset by resolution and re-annotated the top 1k highest-resolution samples. This led to significant improvements, with LLaVA-OV-7B increasing from 55.3 to 56.8. Based on these results, we believe increasing the number of high-resolution samples will improve performance on such tasks. We plan to continue adding more high-resolution data in the future.
>
>
> ----
>
> **Concern 2: Dynamic Reward Scaling technique is a relatively incremental advancement over DPO methods. There are already similar ideas proposed for reward model training [1].**
>
> At first, the reference [1] in the reviewer's question is missing.
>
> In fact, we discuss the differences between Dynamic Reward Scaling, DPO, and its improved versions in line 304 of the original paper. Overall, there are three key distinctions:
>
> 1. Novelty: We are the first to propose the dynamic beta adjustment framework for MLLMs.
>
> 2. Methodological Advance: We demonstrate that instance-level beta tuning is viable with a robust reward model, contrary to prior beliefs [1].
>
> 3. Empirical Gains: Our approach outperforms existing methods (Figure 11(a)).
>
> [1] Beta-DPO: Direct preference optimization with dynamic beta.
>
> Additionally, we have already elaborated on the differences between similar reward model training methods and our approach in line 261. We conclude the main content as follows: In the MLLM community, there is no unified framework for designing reward models. Some approaches use traditional reward models with limited interpretability, while others rely on LLMs for ranking, which often leads to high variance. Additionally, other works focus on improving the reliability of model-generated critiques, but with a different goal. Our study is the first to explore how MLLMs can effectively leverage human annotations to enhance both interpretability and scoring ability.
>
> For a detailed discussion, please refer to Appendix F (Comparison to Existing Methods on Beta Adjustment in LLMs and MLLMs) and Appendix E (Discussion of MM-RLHF-Reward Model).
>
> ----
>
> **Concern 3: This paper did not compare itself or discussed some previous RLHF methods or datasets for multimodal LLM training, like MIA-DPO and PPLLaVA, LLaVA-OneVision-Chat**
>
> PPLLaVA directly uses DPO loss without any improvements, so a direct comparison is not feasible. For a more detailed answer to the reviewer’s question, please refer to this link (https://anonymous.4open.science/r/mm-rlhf-rebuttal-BE17). We compared our approach to various baselines, including LLaVA-Critic, beta-DPO, SIMPO, and MPO, and found that existing methods showed limited gains on our high-quality preference dataset. Additionally, we compared DPO training results across multiple datasets (e.g., VL Feedback, RLAIF, LLaVA-Critic, MPO-Data), and the results demonstrated that MM-RLHF provides more comprehensive and significant performance improvements compared to existing datasets.
>
> ---
>
> **Concern 4: Is all possible pairs necessary? and What does “MM-RLHF” refer to—SFT on high-rated examples or the reward model’s output?**
>
> Please refer to this link (https://anonymous.4open.science/r/mm-rlhf-rebuttal-BE17) for  **General Response 2. Comparison of SFT Baselines and DPO Sample Selection Strategies.**
>
> ---
>
> **Concern 5: Figures would benefit from error bars or statistical significance indicators.**
>
> Thank you for the suggestion. Since we use fixed seeds during training and temperature=0 for deterministic generation during evaluation, performance variance is generally small. However, we agree this is a valuable improvement. Due to the time limit, we will report the mean and standard deviation across three runs with different seeds in the final version.
>
> ---
>
> **Concern 6: Why train the MM-RLHF reward model if it’s not used for PPO or other RL methods?**
>
> The reward model is essential in our MM-DPO method for computing reward margins.
> We did experiment with PPO as well, but similar to the observations in Concern 3 about multi-stage DPO, PPO requires online sampling and fine-grained hyperparameter tuning. We only observed improvements in relatively simple dialogue tasks with a 7B LLaVA-ov model, and training stability was a challenge. In contrast, DPO-based methods offer more robust performance, especially given the high-quality responses (e.g., from Qwen2-VL-72B) in the MM-RLHF dataset, making them a more practical and effective choice.

---

### Official Review · Reviewer_QAMA · 2025-03-14

**Overall Recommendation:** 3

**Summary:**

This work introduces MM-RLHF, a new dataset for fine-tuning multimodal large language models (MLLMs) with human preference. The data samples are collected from diverse sources and carefully annotated by expert human annotators. Based on this new dataset, a reward model training framework is developed, which generates text critiques before scoring. Along with the newly proposed dynamic reward scaling technique, the method improves several MLLMs' capabilities in a wide range of benchmarks.

**Claims And Evidence:**

No concerns.

**Essential References Not Discussed:**

No concerns.

**Experimental Designs Or Analyses:**

No concerns.

**Methods And Evaluation Criteria:**

- When constructing the dataset, this work does not seem to explicitly incorporate a mechanism that prevents test data leakage/contamination. In other words, some test questions and/or images may be included in the training data of MM-RLHF, and model performance is improved on such test samples.

- The human preference annotations (as detailed in Appendix B) can be subjective and vary among annotators. It is unclear how to improve the annotation consistency and reduce the biases. This work does not include an evaluation of annotation consistency of different human annotators.

- Although the data samples are re-sampled to create a balanced mixture of different topics, there seems no clear evidence that the dataset has sufficient geographical diversity.

**Other Comments Or Suggestions:**

- Please improve the resolution of Figure 1.

**Other Strengths And Weaknesses:**

No more concerns.

**Questions For Authors:**

No more questions.

**Relation To Broader Scientific Literature:**

This work introduces a new dataset for training MLLMs with human preference, which is a great contribution to the MLLM community. Empirical results show improved performance brought by this dataset. However, there are a few remaining concerns regarding the dataset construction, which should be addressed before publication.

**Theoretical Claims:**

This work does not include theoretical claims.

---

> ### Author Rebuttal · Authors · 2025-03-28
>
> Thank you for your time and for acknowledging our work. We will address each of your concerns below:
>
> ----
>
> **Concern 1: This work does not seem to explicitly incorporate a mechanism that prevents test data leakage/contamination**
>
> Thank you for raising this important issue. In our work, we have implemented several measures to minimize the risk of leakage between the training and test datasets:
>
> 1. **Strict separation between training and test data**: During the data sampling process, we manually ensure that the selected samples are from the training set and have no overlap with the evaluation subset.
> 2. **Experimental validation**: We tested our results on more than twenty benchmarks, and observed consistent improvements across domains, with no signs of overfitting in any specific domain. The safety-related data showed the largest improvements because the model had minimal exposure to safety-related issues during pretraining. However, our safety data was newly constructed, clearly distinct from the benchmarks used.
>
> -----
>
> **Concern 2: This work does not include an evaluation of the annotation consistency of different human annotators.**
>
> In fact, our annotation process involves multiple rounds of interactive validation (at least two rounds). We will include the following details in the main text:
>
> 1. **Clear annotation guidelines and training**: As shown in Appendix B, we provided detailed annotation guidelines and training for annotators, ensuring that they could consistently understand and execute the annotation tasks. This helps reduce annotation inconsistency among different annotators.
> 2. **Annotation review and iteration**: To further improve consistency, we implemented a multi-step review process. The first annotator performs the annotations, then another annotator reviews them to ensure agreement. In cases of inconsistency, a third annotator is introduced, and the final decision is made by selecting the most suitable annotations.
>
> ----
>
> **Concern 3: There seems to be no clear evidence that the dataset has sufficient geographical diversity.**
>
> This is an interesting point. In Figure 3, we show the task richness of the dataset, and the images involved may contain buildings or natural landscapes from various geographical locations around the world.
>
> However, since most existing public datasets are in English, our mm-rlhf model is also affected by this issue. The majority of the scenarios are still based on English-language contexts. As the reviewer pointed out, there may be a lack of sufficient geographical diversity. We have recognized the importance of geographical diversity and are actively working on collecting multimodal data from diverse geographical locations and languages, including Chinese (Mandarin), French (French), and other commonly spoken languages, to further enhance the geographical diversity of our dataset.
>
> ----
>
> **Concern 4: Please improve the resolution of Figure 1.**
>
> Thank you for the suggestion! We have updated the original image to a PDF format to avoid the issue of low resolution.

---

> > ### Comment · Reviewer_QAMA · 2025-04-04
> >
> > The authors' response is greatly appreciated. I will adjust the rating as "weak accept." Regarding the mechanism for preventing data leakage, a better strategy could be, for example, measuring the similarity between images in the evaluation benchmarks and the training samples (in some embedding space), and investigate the ones that are very similar. Some concerns raised by other reviewers are also valid.

---

> > > ### Author Response · Authors · 2025-04-07
> > >
> > > Thank you for your prompt response !
> > >
> > > We fully agree with your suggestion that measuring the similarity between images in the evaluation benchmark and the training samples could be an effective strategy to prevent potential data leakage. However, with our initial sample of 100k image samples and over 20 evaluation benchmarks, computing the similarity across all images becomes extremely challenging. Additionally, due to the diversity of the benchmarks, it is difficult to pre-identify which benchmarks might be susceptible to data leakage. Therefore, for large-scale datasets such as LLaVA-OV-Image (with 3.5 million samples), there is currently no highly efficient strategy to prevent data leakage. We are actively conducting related experiments to further filter the training data, but it is unlikely that we will be able to complete these before the response deadline.
> > >
> > > In the case of the MM-RLHF-reward benchmark, we quickly conducted a relevant experiment where we removed the training images most similar to the top 100 images in the benchmark, along with their corresponding questions. The performance change was negligible, with an observed difference of less than 1%, as this subset of data only accounted for a very small portion of the training set. No evidence of data leakage was found during this process.
> > >
> > > If you have any further questions, please feel free to ask.

---

### Decision · Program_Chairs · 2025-05-01

**Decision:**

Accept (poster)

**Comment:**

This paper introduces MM-RLHF, a large-scale multimodal human preference dataset and two methodological contributions—critique-based reward modeling and dynamic reward scaling for improved DPO training—aimed at enhancing alignment of multimodal LLMs. The work addresses a timely and important problem, offering strong empirical results across 27 benchmarks in areas such as safety, reasoning, and hallucination control. While some reviewers raised concerns about the necessity of the proposed methods and the potential for benchmark bias, the authors provided a thorough rebuttal with clarifications, new experiments, and evidence of careful train/test separation. As suggested by the reviewer, the paper would benefit from restructuring and highlighting key content in the main text for clarity. With the changes permitted, I recommend acceptance.